# A multi-functional role for the MCM8/9 helicase complex in maintaining fork integrity during replication stress

Wezley C. Griffin[1,2], David R. McKinzey[1], Kathleen N. Klinzing ⬤[1], Rithvik Baratam[1], Achini Eliyapura[1] & Michael A. Trakselis ⬤[1] ✉

The minichromosome maintenance (MCM) 8/9 helicase is a AAA$^+$ complex involved in DNA replication-associated repair. Despite high sequence homology to the MCM2-7 helicase, a precise cellular role for MCM8/9 has remained elusive. We have interrogated the DNA synthesis ability and replication fork stability in cells lacking MCM8 or 9 and find that there is a functional partitioning of MCM8/9 activity between promoting replication fork progression and protecting persistently stalled forks. The helicase function of MCM8/9 aids in normal replication fork progression, but upon persistent stalling, MCM8/9 directs additional downstream stabilizers, including BRCA1 and Rad51, to protect forks from excessive degradation. Loss of MCM8 or 9 slows the overall replication rate and allows for excessive nascent strand degradation, detectable by increased markers of genomic damage. This evidence defines multifunctional roles for MCM8/9 in promoting normal replication fork progression and genome integrity following stress.

Accurate genomic duplication during S-phase is vital such that each daughter cell is guaranteed a copy of the complete, unadulterated genome. Several thousand replication complexes are licensed and fired with temporal and spatial precision to ensure ephemeral but complete DNA replication[1,2]. The replication machinery often encounters a variety of challenges, including DNA template damage, DNA secondary structure, or DNA-protein blocks[3,4]. These challenges often stall replication forks, either temporarily or more persistently, and if not rescued or restarted by a variety of DNA damage responses (DDR) can collapse into DNA double-strand breaks (DSBs). Such breaks are hallmarks of chromosomal instability that contribute to cancer development, aging, and infertility[5–7].

Fortunately, cells have evolved several failsafe mechanisms to thwart the deleterious outcomes of replication fork stalling and collapse through activation of fork protection pathways[8]. These pathways are coordinated by the ATR kinase, which signals a variety of downstream stress responses that inhibit cell cycle progression, suppress late origin firing, and ensure stabilization and recovery of stalled or reversed replication forks[9]. Replication fork reversal continues to gain

support as a general defense mechanism to protect stressed forks and prevent fork collapse[10–12]. The fork reversal/restart mechanism can be sub-divided into three basic steps: (1) SNF2 enzyme-mediated annealing of newly synthesized and parental DNA strands to form a regressed arm and a four-way DNA junction or 'chicken foot' structure, (2) removal of damage or replication block, and (3) re-installation and restart of the replication complex. In addition, many proteins important for homologous recombination (HR) repair of DSBs (e.g., BRCA1/2, RAD51, MRE11, etc.) moonlight at stalled or reversed forks to prevent genomic instability through fork protection/restart or recombination[13].

The minichromosome maintenance (MCM) 8 and 9 are ATPases associated with a variety of cellular activities (AAA$^+$) and are homologs within the MCM family of proteins. While MCM2-7 forms the core of the replicative helicase, MCM8 and 9 form a discrete heterohexameric helicase complex implicated in HR-mediate repair[14]. Studies have linked the loss of MCM8 or 9 to primary ovarian failure (POF), infertility[15–17], and cancer[18], with more than 400 different mutations in both MCM8 and 9 cataloged in genome databases[14]. Many of these

[1]Department of Chemistry and Biochemistry, Baylor University, Waco, TX 76706, USA. [2]St. Jude Children's Research Hospital, Memphis, TN 38105, USA. ✉e-mail: michael_trakselis@baylor.edu

reports show a direct link between a functional MCM8/9 complex and successful meiotic or mitotic HR. However, these mutations lack sufficient characterization of the molecular and cellular effects that contribute to disease initiation and progression. Indeed, both mice and humans with non-functional MCM8/9 display reproductive system abnormalities including infertility, sex-specific tumor formation, sensitivity to DNA damaging agents, and defects in HR processing[19,20]. Furthermore, loss of MCM8/9 impairs HR-mediated fork rescue due to decreased recruitment of the MRN helicase/nuclease complex, RAD51 recombinase, and RPA single-stranded (ss-) DNA binding protein after cisplatin (cis-Pt) treatment[21].

Despite a high sequence homology to MCM2-7, a precise function of the MCM8/9 at the DNA replication fork has remained enigmatic. Although early reports debated the role for MCM8/9 at prereplication complex (preRC) assembly[22] or during active replication[23–26], the focus quickly turned toward MCM8/9's participation in HR processes. However, replication forks in cells lacking MCM8 or 9 stalled or collapsed nearly 2-fold more than control cells, suggesting loss of MCM8/9 sensitizes forks to replication stress[19]. In addition, when MCM2 is rapidly degraded, MCM8/9 can fill in and allow for DNA-dependent synthesis, albeit at a significantly slower overall rate[27]. Furthermore, isolation of proteins on nascent DNA, or iPOND, supports the presence of MCM8/9 at active replisomes at a similar level to other bona fide replication proteins[28]. Together, this suggests that MCM8/9 actively contributes to genomic integrity by promoting replisome progression through the stabilization and protection of DNA replication forks during active elongation.

Here, we report a role for the MCM8/9 complex in maintaining replication fork stability during fork progression, stalling, and reversal. By integrating single-molecule DNA fiber and neutral comet assays with flow cytometry and immunofluorescence analyses, we show that MCM8/9 knockout (KO) cell lines exhibit reduced rates of DNA synthesis, delayed cell cycle progression, and increased markers of genomic instability because of reduced replication fork protection. Collectively, our data support a multi-functional model, whereby the helicase domain of MCM8/9 antagonizes BRCA1-dependent fork reversal, stabilization, and processing to promote normal fork progression. However, upon excessive stalling, MCM8/9 recruits RAD51 through a BRCv motif in the C-terminal extension of MCM9 to protect and reverse stalled forks. These results confirm a direct role for MCM8/9 in maintaining genomic integrity by stabilizing the active replication fork and facilitating protection of a persistently stalled fork.

## Results

### Knockout of the MCM8/9 complex slows DNA synthesis progression and sensitizes cells to replication-induced DNA damage

Recently, it has been shown that multiple proteins with established roles in HR and repair also have activities in maintaining replication fork progression and integrity[29]. Since MCM8/9 are homologous to the MCM2-7 replicative helicase complex and are involved in HR repair, we hypothesized that MCM8/9 may also be involved in maintaining fork integrity during replication. Previous observations have suggested that cells lacking MCM8 or 9 exhibit reduced growth rates[19,27], which could be explained by compromised fork progression or stability. To assess this possibility, we created knockouts of MCM8 (8^KO) or MCM9 (9^KO) using CRISPR/Cas9. This approach yielded an absence of MCM8 in 8^KO cells and a reduction of MCM8 in 9^KO cells (Fig. 1a); while in both the 8^KO and 9^KO cells, there is a near complete knockout of MCM9 (Fig. 1b). It was previously shown that the stability of MCM8 and MCM9 were dependent on each other, as knockout or knockdown of one also reduced or eliminated levels of the other[19,24,30].

Both 8^KO or 9^KO cells grew in culture at a qualitatively slower rate than WT cells. To directly quantify S-phase progression, we performed cell synchronization experiments with a double thymidine block at the G1-S phase boundary. After release, cell cycle progression through S-phase was monitored by fluorescence-activated cell sorting (FACS) (Fig. 1c) and gating cell populations by propidium iodide signal into S, G2/M, and G1 (Fig. 1d and Supplementary Fig. 1). The 2–3 h delay through S-phase for 8^KO or 9^KO cells compared to wild-type translates to an overall delay in cell division (G2/M) and continues through the next G1 phase.

To monitor global apparent DNA synthesis rates in the absence of exogenous damage, we first utilized DNA fiber analysis with single pulse CldU at different times for two separate clones of 8^KO or 9^KO compared to parental cells (Supplementary Figs. 2a, 3a). Mean CldU track length values were then plotted as a function of time and fit to a simple linear regression to obtain apparent DNA synthesis rates (Supplementary Figs. 2b, 3b). The CldU track length in wild-type (WT) 293T cells increased at a rate of ~0.21 μm per minute, which corresponds to a DNA synthesis rate of 10.4 base pairs per second. As expected, both 8^KO and 9^KO cells exhibited 2–3-fold reduced CldU track length rates of 0.04 and 0.03 μm per minute, which correspond to DNA synthesis rates of 2.0 and 1.7 base pairs per second, respectively. To further validate these cell lines and show that MCM8 and MCM9 are directly aiding in DNA synthesis processes, transfection of MCM8 or MCM9 back into their respective knockout lines restored the DNA synthesis rates (Supplementary Fig. 4) with transfection efficiencies of 80% (Supplementary Fig. 5).

DNA fiber experiments with single modified nucleotide pulses can be complicated to interpret, as they cannot adequately separate fork speed from conflicting DNA repair processes and further origin activation events, especially at later timepoints. To better distinguish replication fork speeds, we turned to a dual labeling approach, where a consistent 30-min CldU pulse is followed by a second variable (30–60 min) IdU pulse to better quantify replication fork progression speed. Only those IdU tracts adjacent to a CldU track are quantified. Here, it is clear that fork progression in 8^KO or 9^KO cells is severely compromised, compared to parental cells, and essentially shows no further fork progression after the second 30-min IdU pulse (Fig. 1e–g). It is likely that replication forks can only proceed for a short period of time (<~60–75 min total) before they become unstable and prone to degradation, requiring DNA repair processing and increased origin activation consistent with some increase in fiber lengths with the single pulse CldU experiments (Supplementary Figs. 2, 3). Together, this provides strong evidence that the MCM8/9 complex aids in normal replication fork progression and that the loss of MCM8/9 likely results in reduced replication fork protection, resulting in genome instability.

Previously, several laboratories, including ours, have shown that MCM8/9 form nuclear foci upon damage, primarily from DNA crosslinking agents such as mitomycin C (MMC), or after direct DSBs induced by ionizing radiation[24,30,31]. While these studies implicate MCM8/9 in HR, there is limited information investigating a possible role during fork progression/stalling. To directly examine whether MCM8/9 are involved in maintaining genomic stability during replication stress, a GFP-tagged MCM9 fusion construct was transfected into WT 293T cells, after which cells were treated with 2 mM hydroxyurea (HU) for 4 h to induce fork stalling, and MCM9 foci formation was monitored (Fig. 2a and Supplementary Fig. 6). Cells treated with HU exhibited more MCM9-dependent foci compared to GFP alone and nontreated controls, indicating that the MCM8/9 complex responds to stressors that induce replication fork stalling.

Replication-associated DNA damage in both 8^KO and 9^KO cells was directly measured and compared to WT cells using a neutral comet assay to detect DSBs (Fig. 2b). Upon treatment of 8^KO and 9^KO with 2 mM HU for 4 h, there is a statistically significant (~2.2- and ~1.7-fold, respectively) increase in tail moment values compared to WT cells. Addition of MMC to 8^KO or 9^KO cells showed a similar trend with a ~1.5- and ~2.1-fold increase in tail moment compared to WT, respectively. To

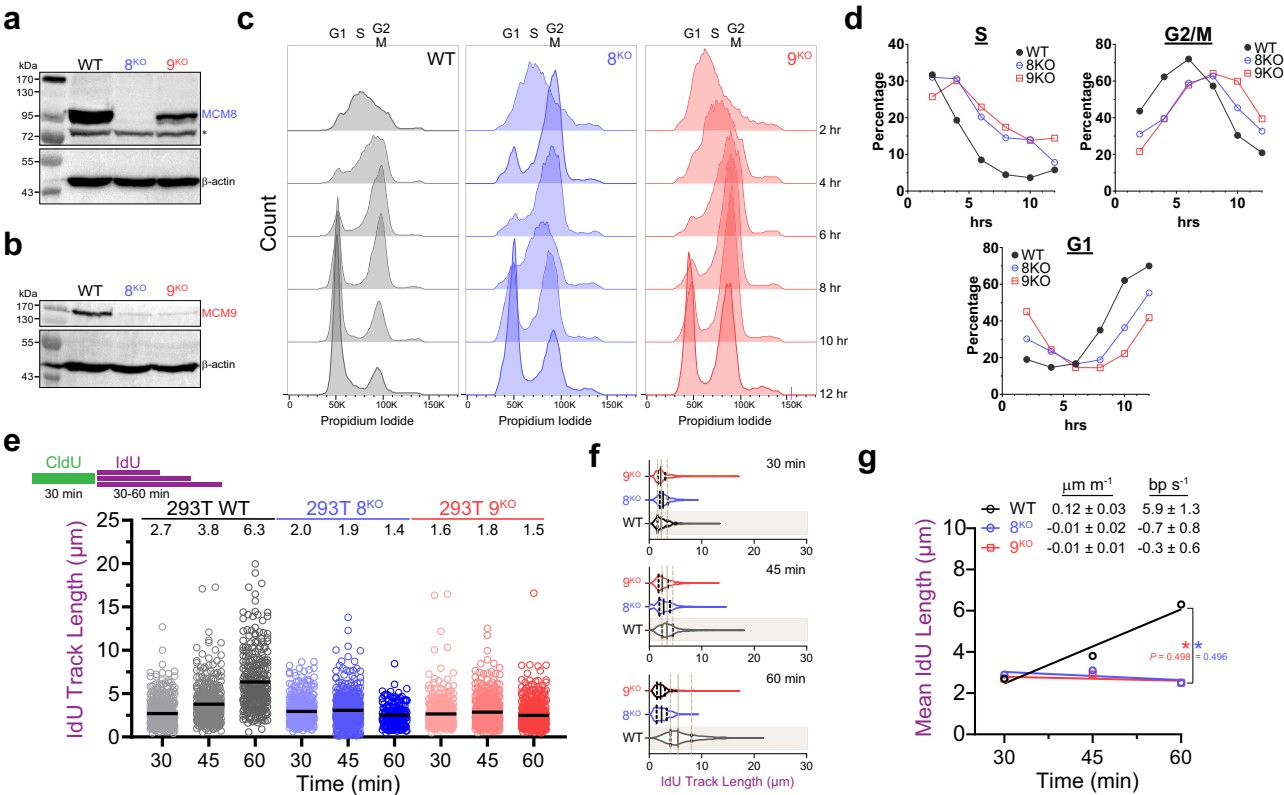

**Fig. 1 | DNA replication rates are reduced in the absence of MCM8/9, leading to delayed cell cycle progression.** Western blots showing the absence of **a** MCM8 or **b** MCM9 in their respective knockout cell lines ($n = 2$ independent blots). B-actin was used as a loading control. *indicates the presence of a nonspecific band. **c** Cells were synchronized with a double thymidine block, released into S-phase, and then the chromosome content was monitored by FACS. **d** After gating and quantification of S, G2/M, and G1 populations, the percentages were plotted as a function of time. **e** Dual labeling DNA fiber assay with a constant 30 min CldU pulse followed by a variable 30–60 min IdU pulse. The IdU track lengths adjacent to a constant CldU track (>400 fibers each) were measured as a readout of replication progression for 293T WT (gray circles, ○), 8$^{KO}$ (8B2 clone, blue circles, ○), 9$^{KO}$ (9D6 clone, red circles, ○) cells. IdU lengths were measured with ImageJ software and the corresponding mean value of each time point are indicated above the plot and with a black bar. **f** Violin plots comparing the IdU track lengths at each time point to the WT values (beige box and lines) with mean (solid line) and quartiles (dashed lines) indicated. **g** Mean IdU track length values were plotted as a function of time to obtain apparent overall replication rates. A two-sided *t* test was used to calculate *P* values between the slopes for either WT and 8$^{KO}$ or 9$^{KO}$ (*$P < 0.05$).

show specificity, comet tail moments are rescued by the over-expression of untagged MCM8 or MCM9 from an IRES2 plasmid after treatment with HU (1.2- vs. 1.5-fold reduction, respectively) or MMC (1.3- vs. 1.9-fold reduction, respectively) (Supplementary Fig. 7).

To further investigate the prevalence of DNA breaks occurring in 8$^{KO}$ or 9$^{KO}$ cells, γH2A.X foci were probed in nontreated or HU-treated cells (Fig. 3a, b and Supplementary Fig. 8). γH2A.X foci are surrogate markers of DNA damage and early effectors of the DSB repair pathway[32,33]. Interestingly, both 8$^{KO}$ or 9$^{KO}$ cells showed significant increases in γH2A.X foci in nontreated cells, consistent with the hypothesis that loss of MCM8/9 results in defective replication that induces genomic stress (Fig. 3c). Nontreated WT cells were essentially void of any γH2A.X foci. This effect was enhanced overall with HU treatment, where significantly more foci were again found in 8$^{KO}$ or 9$^{KO}$ cells compared to WT cells (Fig. 3d). These results indicate that cells lacking MCM8/9 are more susceptible to DNA damage-inducing events, likely initiated by reduced fork stability during replication that results in more rampantly reversed forks mimicking a DSB end recognized by γH2A.X.

## MCM8/9 maintains replication fork integrity during stress and reversal

As MCM8/9 appears to be involved in aiding replication fork progression and maintaining genomic integrity, we hypothesized that MCM8/9 may act in a similar manner as other HR proteins to stabilize stalled or reversed replication forks and protect against

nascent strand degradation (NSD)[34–36]. To examine this possibility, we measured replication fork stability in WT, 8$^{KO}$, and 9$^{KO}$ cells by DNA fiber analysis, examining whether degradation of the nascent strand occurs (Fig. 4a). Interestingly, in untreated conditions, both 8$^{KO}$ and 9$^{KO}$ cells exhibit a statistically significant reduction in the median IdU/CldU ratio value compared to WT (Fig. 4b, compare plots 1, 3, and 7 with open circles, and in Supplementary Fig. 9a, i), suggesting a defect in replication fork protection upon loss of the MCM8/9 complex. This reduction in median IdU/CldU values was more pronounced in the presence of 2 mM HU, which stimulates more persistent replication stress and initiates fork reversal (Fig. 4b, compare plots 2, 5, and 9 with filled circles, and in Supplementary Fig. 9a, ii), indicating that loss of MCM8/9 further sensitizes replication forks to degradation following stress. Furthermore, transfection of WT MCM8 or MCM9 constructs into the respective KO cells partially restores replication fork stability following 2 mM HU treatment, as indicated by the increase in median IdU/CldU value compared to GFP alone transfected controls (Fig. 4b, compare plots 6 vs. 5 and 10 vs. 9, and in Supplementary Fig. 9a, iii–iv). We note that DNA fiber measurements and quantifications for transfection of GFP into 9$^{KO}$ cells (Fig. 4b, plots 11 and 12) are nearly identical to that of nontreated and HU-treated 9$^{KO}$ cells (Fig. 4b, plots 7 and 9), highlighting the reproducibility of our methods and providing confidence for fiber quantification throughout. Furthermore, we repeated these DNA fiber measurements in the absence and

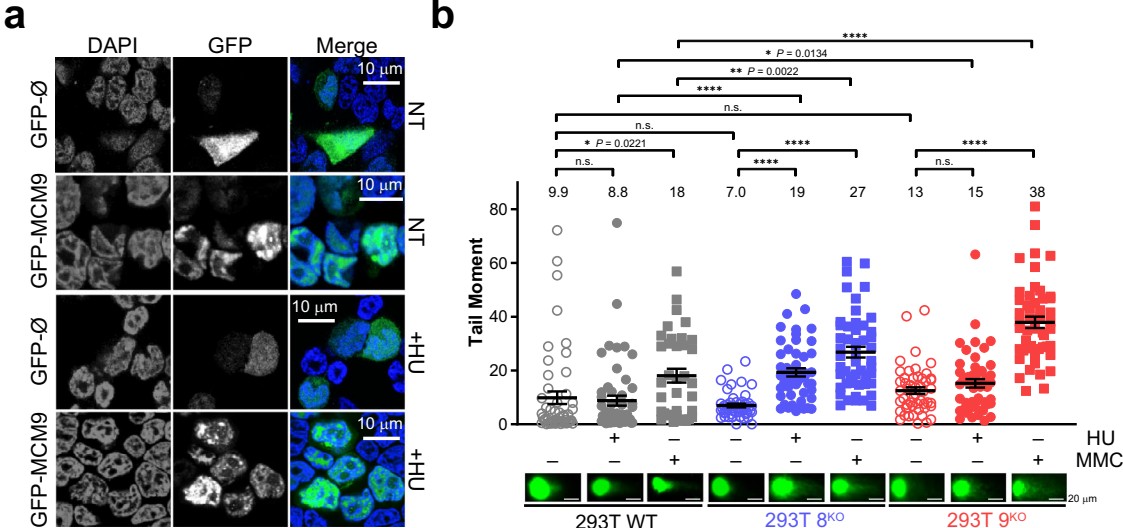

**Fig. 2 | MCM8/9 protect forks from replication-induced DNA breaks. a** 293T cells were transfected with GFP-MCM9 fusion construct or GFP alone (Ø), nontreated (NT) or treated with 2 mM HU for 4 h (+HU) and imaged by confocal microscopy ($n = 2$ independent experiments). Scale bars represent 10 μm. Larger representative views of microscopy images of cells are shown in Supplementary Fig. 6. **b** Quantification of the tail moment (>50 cells for each condition) in a neutral comet assay for 293T WT (gray), 8^KO (blue), or 9^KO (red) cells (○) treated with 2 mM HU for 4 h (●) or 3 μM MMC for 6 h (■). Mean values are listed above the plots and with a black line embedded in the data, and error bars represent the standard error of the mean (SEM) (n.s. nonsignificant, *$P < 0.05$, **$P < 0.01$, ***$P < 0.0001$, by paired two-sided $t$ test). Representative comets are shown below the graph for each condition with included white scale bars (20 μm).

presence of HU for these clones as well as two other MCM8^KO or MCM9^KO clones and can show similar fork instability profiles and trends (Supplementary Fig. 10).

## MCM8/9 stabilizes stalled forks and protect from nucleolytic degradation

Several nucleases (MRE11, EXO1, and DNA2) have reported activities in processing reversed replication forks to initiate fork recovery and restart[37–39]. However, when excessive fork stalling occurs or when fork protectors are deficient or absent, dysregulated nucleolytic degradation of the nascent strand by these nucleases is hypothesized to be a source of genomic instability. Based on these previous observations, we hypothesized that the fork instability in both 8^KO and 9^KO cells was a result of aberrant or excessive NSD and that inhibition or knockdown of these nucleases might restore fork stability.

After knockdown of MRE11, EXO1, or DNA2 by siRNA in both 8^KO and 9^KO cells, fork integrity was examined by DNA fiber analysis (Fig. 4c–f). Knockdown of MRE11 in both 8^KO or 9^KO cell lines restored the minor fork instability in nontreated cells to WT levels (Fig. 4c, plots 3 and 5 with open circles, and in Supplementary Fig. 9b, i, compared with plots 3 and 7 in Fig. 4b, and in Supplementary Fig. 9a, i). Interestingly, there was a minor but significant decrease in fork protection in WT cells treated with 2 mM HU (Fig. 4c, plots 1 and 2), which highlights the activity of multiple nucleases involved in reversed fork degradation. The addition of 10 μM Mirin (a MRE11 inhibitor) also restored replication fork protection in 8^KO and 9^KO cells treated with HU (Fig. 4d, plots 4 and 6 with filled circles, and in Supplementary Fig. 9c, ii, compared with plots 5 and 9 in Fig. 4c) but not in the nontreated conditions (Fig. 4d, plots 3 and 5 with open circles, Supplementary Fig. 9c, i, compared with plots 3 and 7 in Fig. 4c) suggesting that alternative forms of fork degradation or controlled nucleolytic processing (i.e., EXO1 or CtIP and DNA2) are still active in the absence of both HU stress and MCM8/9, consistent with the hypothesis that multiple mechanisms of fork processing for restart are utilized by the cell[40]. It is also possible that as Mirin primarily inhibits the exonuclease activity of MRE11[41], the remaining endonuclease activity of MRE11 may be responsible for fork instability in the nontreated conditions.

Knockdown of EXO1 (Fig. 4e and Supplementary Fig. 9d) showed a similar trend in fork protection restoration comparable to that observed for siMRE11 (Fig. 4c), where 8^KO and 9^KO cell lines have restored fork protection in nontreated and HU treated conditions (compare Fig. 4e with plots 3 and 7 and 5 and 9 in Fig. 4b). This is consistent with these nucleases working in concert to process stalled or reversed replication forks. Last, knockdown of DNA2 restored replication fork protection across all conditions examined including WT (Fig. 4f and Supplementary Fig. 9e), implicating DNA2 as an additional nuclease that can process or degrade replication forks through multiple mechanisms. For example, it has been hypothesized that DNA2 can stably associate with replication forks and counteract fork reversal by degrading the nascent strands during regression[39]. The rescue of the minor fork degradation phenotype seen after knockdown of DNA2 in WT cells treated with HU (compares plots 1 and 2 in Fig. 4F with plots 1 and 2 in Fig. 4c, e) would suggest that DNA2 may be the nuclease responsible for the minor fork degradation seen in cells knocked down for Mre11 or Exo1, consistent with previous observations[40]. In addition, CtIP has been proposed as an important regulator of DNA2 activity in preventing excessive fork degradation after stalling with HU[42]. Overall, the restorative effect of knocking down these nucleases in the absence of MCM8/9 supports the conclusion that MCM8/9 has a general protective role in preventing multifaceted nucleolytic degradation of transiently and more severely stalled replication forks.

Several SNF2 helicase-like ATPase remodeling enzymes (SMAR-CAL1, HLTF, ZRANB3) catalyze replication fork reversal upon stalling. To examine whether MCM8/9 actively stabilizes forks reversed by these enzymes, SMARCAL1 and HLTF were knocked down separately by siRNA transfection in 8^KO or 9^KO cells, and NSD was measured by DNA fiber analysis. Transfection with siSMARCAL1 rescued the minor decrease median IdU/CldU ratio values in both nontreated 8^KO and 9^KO to WT levels (Fig. 5a, compare plots 3 and 5 with open circles, and Supplementary Fig. 9f, i, with plots 3 and 7 in Fig. 4b). This rescue in median IdU/CldU ratio values was also observed after treatment with 2 mM HU (Fig. 5a, compare plots 4 and 6 with filled circles and Supplementary Fig. 9f, ii, with plots 5 and 9 in Fig. 4b). These data suggest

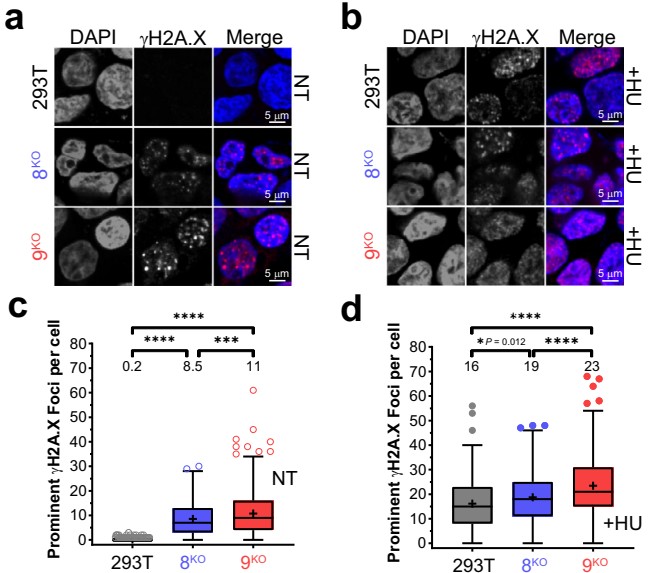

**Fig. 3 | MCM8/9 protects cells from γH2A.X markers of DNA damage during normal replication and HU-induced stalling.** 293T (gray), 8^KO (blue), or 9^KO (red) were examined for either **a** native γH2A.X foci in nontreated (NT) cells or **b** after treatment with 2 mM HU for 4 h (+HU) and then imaged by confocal microscopy (n = 3 independent experiments). Scale bars represent 5 μm. Larger representative views of microscopy images of cells are shown in Supplementary Fig. 8. Prominent γH2A.X foci were quantified per cell for more than 230 nuclei using Image J and plotted as a Tukey box and whiskers for **c** NT or **d** HU treated cells. Mean values are listed above the plots and with a + embedded in the data. Box boundaries represent the 1st and 3rd interquartile range (IQR); the line represents the median; the upper and lower whiskers are designated at a data point ≤ (±1.5× IQR); and individual data points outside the whiskers are shown. A Mann–Whitney two-sided U test was used to calculate P values (*P < 0.05, ***P < 0.001, ****P < 0.0001).

that, in the absence of MCM8/9, replication fork protection is compromised and prone to rampant NSD, following prevalent SMARCAL1 fork reversal.

Conversely, addition of siHLTF to both nontreated 8^KO and 9^KO cells reduced IdU/CldU ratios to levels analogous to that observed for 2 mM HU treated conditions (Fig. 5b, compare open and filled circles and Supplementary Fig. 9g), and unlike for siSMARCAL1, fiber ratios were not rescued. No significant change in fork protection was observed in the WT cells. The reduction in replication fork protection following siHLTF in both nontreated and treated conditions suggests that MCM8/9 function in a complementary but non-overlapping replication fork protection pathway. Indeed, HLTF has been reported to protect replication forks via alternative mechanisms[43]. Instead, MCM8/9 likely functions to stabilize reversed replication forks contained within the SMARCAL1 axis.

**MUS81 robustly cleaves stalled forks in the absence of MCM8/9**
We next wanted to address if MCM8/9 are directly involved in replication fork restart. In this experiment, we treated cells with CldU for 30 min followed by co-treatment with 2 mM HU with 10 μM Mirin (to prevent nucleolytic degradation of CldU tracts) followed by release from HU and incubation in IdU for 30 min to allow stalled replication forks to restart. Both 8^KO and 9^KO cells did not efficiently restart replication forks compared to WT (Fig. 5c, filled plain circles, plots 1 vs. 2 and 4). Interestingly, transfection of GFP-tagged MCM8 or 9 into their respective KO cells also did not efficiently restore replication fork restart (Fig. 5c, green outlined circles, plots 3 and 5 vs. 2), suggesting that the MCM8/9 complex is not directly involved in fork restart activities, or it promotes alternative mechanisms of replication fork restart (such as HR-mediated). Transfection of the GFP only control in

9^KO cells did not allow for efficient restart, as expected (Fig. 5c, green filled circles, plot 6).

Persistent replication fork stalling and inefficient restart often leads to MUS81-mediated cleavage to initiate HR-mediated repair or fork restart[44–46]. To investigate whether forks stalled in MCM8/9 KO cells are cleaved by MUS81, we knocked down MUS81 using siRNA and examined NSD by DNA fiber analysis. Knockdown of MUS81 did not restore replication fork protection in either nontreated 8^KO or 9^KO cells (Fig. 5d, blue or red open circles, plots 3 and 5 compared with plots 3 and 7 in Fig. 4b). However, siMus81 in 8^KO and 9^KO treated with 2 mM HU restored replication fork protection to levels observed in the nontreated conditions (Fig. 5d, compare blue and red open and closed circles, plots 3 vs. 4 and 5 vs. 6, and Supplementary Fig. 9h). It is known that replication forks are minimally processed by nucleases such as MRE11 prior to generating substrates amenable to MUS81 cleavage[37]. Our data support a model in which fork protection is not completely restored in nontreated cells depleted for MUS81, as nucleases are still present to minimally process reversed forks. However, when forks are persistently stalled with HU, fork protection is restored to basal levels in 8^KO or 9^KO cells when MUS81 is knocked down (Fig. 5d, compare closed circles, plots 4 and 6 with those in Fig. 4b, plots 5 and 9), implicating MUS81 as the endonuclease responsible for cleaving stalled forks leading to DSBs detected above in 8^KO or 9^KO cells (Figs. 2b, 3).

## MCM8/9 counteracts and restricts BRCA1's role in fork protection

During HR, BRCA1 supports end resection to generate a 3′ overhang, recruits BRCA2 to the site of damage, and aids in loading (w/BRCA2) of RAD51 onto single-stranded DNA[8,47], resulting in protection of nascent DNA strands from degradation by the nuclease MRE11[37]. Fork protection can be restored in BRCA1 deficient cells through inhibition of any one of the SNF2 fork reversal enzymes: SMARCAL1, HLTF, or ZRANB3[34] or nucleases: MRE11, DNA2, MUS81, and SLX4-ERCC1[48]. Similarly, we wondered whether knockdown of BRCA1 could restore fork protection in MCM8/9 knockout (KO) cells.

DNA fibers were used to examine the role of MCM8/9 in stabilizing HU-stalled forks in the absence of BRCA1. As expected, siBRCA1 reduced median IdU/CldU ratios in HU-stalled WT 293T cells (Fig. 6a, plots 1 and 2). However, fork protection was restored when BRCA1 was knocked down in 8^KO or 9^KO cell lines treated with HU (Fig. 6a, plots 2 vs. 4 or 6 and Supplementary Fig. 9i, compared with plots 5 and 9 in Fig. 4b where BRCA1 is present). Thus, it appears that stabilization of stalled replication forks is compromised in the absence of either BRCA1 or MCM8/9 but is restored when both are absent. This may be explained by the inability to form reversed (unprotected) forks when both BRCA1 and MCM8/9 are absent, as it is our hypothesis that MCM8/9 is required to facilitate this pathway switch from fork progression to reversal/protection. These results emphasize a non-redundant role for MCM8/9 and BRCA1 in maintaining replication fork protection, placing them in the same pathway.

To investigate the dependence and temporal recruitment of BRCA1 in relation to MCM8/9, cells were transfected with GFP-MCM8 or GFP-MCM9 and BRCA1 foci were counted in NT and HU or MMC-treated cells (Fig. 6b–d and Supplementary Fig. 11). Interestingly, the presence or overexpression of MCM8 or MCM9 repressed the formation of BRCA1 foci in treated cells. This was evident in HU-treated cells with significant reduction in BRCA1 foci in all cell lines except 8^KO which was reduced but just outside the 95% confidence level. The effect was even more pronounced in MMC-treated cells, with a significant reduction in BRCA1 foci in all cell lines. This effect was also visually apparent in cells transfected with GFP-MCM8 or GFP-MCM9, where there was a void in BRCA1 foci and signal, unlike in untransfected cells (Fig. 6b and Supplementary Fig. 11). Therefore, MCM8/9 likely acts to antagonize BRCA1-mediated fork processing/stabilization during fork reversal to maintain replication fork protection during

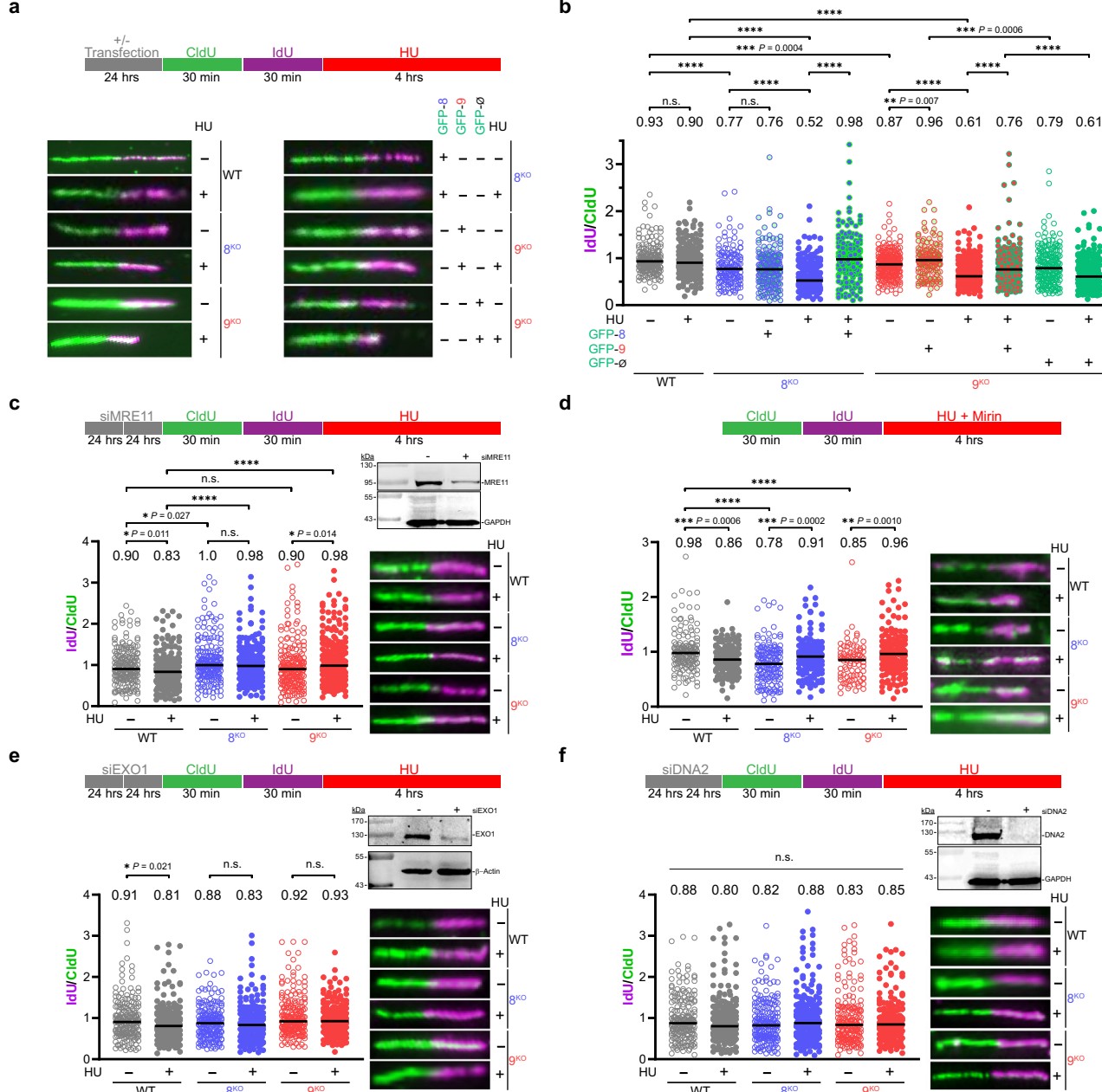

**Fig. 4 | Replication forks are unstable in the absence of MCM8/9 and degraded by a combination of nucleases. a** Cells were labeled with CldU followed by IdU for the indicated time intervals followed by 2 mM HU for 4 h. Experiments containing plasmids for GFP-MCM8, GFP-MCM9, or GFP alone (Ø) were transfected for 48 h prior. DNA was spread by gravity to measure NSD and representative fibers are shown. **b** IdU and CldU lengths for 293T WT (gray), 8^{KO} (blue), or 9^{KO} (red) cells were measured using ImageJ and the corresponding median ratios reported at the top of the plots and by a black line embedded in the data. Open circles (○) represent nontreated (NT) conditions, while solid closed circles (●) represent HU-treated cells. Green circle outlines (○) represent transfected cells. Cell lines were transfected with siRNA twice for 24 h each to knockdown **c** Mre11, **d** treated with 10 μM mirin, or knockdown of **e** Exo1 or **f** Dna2 and DNA fibers processed as above. siRNA knockdown was verified by western blot. Greater than 150 fibers were measured for each condition. A Mann–Whitney two-sided U test was used to calculate *P* values (n.s. nonsignificant, *P < 0.05, **P < 0.01, ***P < 0.001, ****P < 0.0001).

normal replisome progression. However, when severe replisome stalls are prevalent, MCM8/9 hands off the fork template for controlled reversal/protection through the BRCA1/2, RAD51, SMARCAL1 nexus, essentially swapping control of the template.

## The BRCv motif in MCM9 and not helicase activity is necessary to maintain fork protection

Previously, we had characterized a BRC variant motif (BRCv) within the C-terminal extension (CTE) of MCM9 (Fig. 7a) that interacted with and recruited RAD51 to sites of MMC-induced DNA damage[31]. Therefore,

we sought to investigate the role of this MCM9-BRCv motif in maintaining fork protection after HU treatment using DNA fiber analysis. Interestingly, in the absence of HU, fork protection is restored in 9^{KO} cells when MCM9(BRCv^-) is transfected (Fig. 7b, compare plots 1 and 3), implying that the MCM8/9 complex on its own provides some stabilizing context to active replisomes, possibly through its helicase activity. This increase in fork protection was equivalent to that of adding WT MCM9 back in 9^{KO} cells (compare with Fig. 4b, plot 8). When 9^{KO} cells were treated with HU and transfected with MCM9(BRCv^-), fork protection is reduced back to basal levels (Fig. 7b, compare

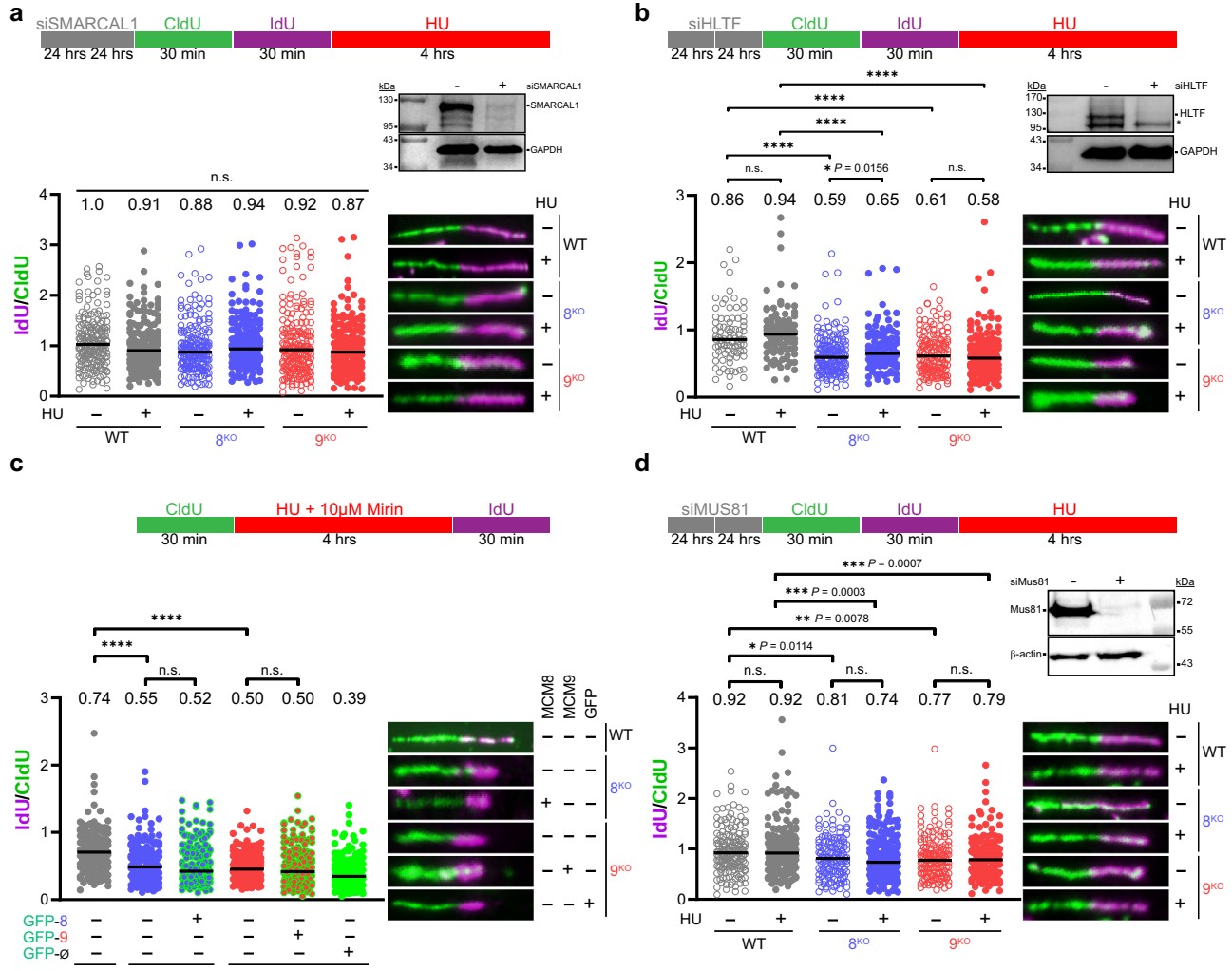

**Fig. 5 | MCM8/9 function to direct SMARCAL1-mediated fork reversal and do not actively participate in stalled fork restart.** 293T WT (gray), 8^KO (blue), or 9^KO (red) cells were transfected with siRNA twice for 24 h each to knockdown **a** SMARCAL1 (>150 fibers) or **b** HLTF (>100 fibers) and verified by western blot (*indicates a nonspecific band on the HLTF blot). Cells were then sequentially incubated with CldU and IdU for the indicated time intervals (○) or followed by 2 mM HU for 4 h (●). DNA was spread by gravity to measure NSD by quantifying fiber lengths using Image J. **c** Cells were treated with 2 mM HU and 10 μM mirin for 4 h to monitor fork restart as flanked by CldU and IdU labeling for indicated time intervals (>150 fibers). Cells were transfected with GFP-MCM8 (green outline, ○), GFP-MCM9 (green outline, ○), or GFP alone (Ø) (solid green, ●) as indicated for 24 h before initiating fork stalling and restart. **d** Cells were transfected with siRNA twice for 24 h each to knockdown MUS81 and then treated as in (**a**) and (**b**) (>140 fibers). IdU and CldU lengths were measured using ImageJ and the corresponding median ratios reported at the top of the plots and by a black line embedded in the data. Representative fibers are shown to the right of the dot plots. A Mann–Whitney two-sided U test was used to calculate *P* values (n.s. nonsignificant, **P* < 0.05, ***P* < 0.01, ****P* < 0.001, *****P* < 0.0001).

plots 2 and 4) and lower than that for adding WT MCM9 (Fig. 4b, plot 10), suggesting that recruitment of RAD51 is required to provide stabilization to more persistent HU reversed forks facilitated by the BRCv motif of MCM9.

Based on this separation of function mutation for MCM9(BRCv⁻) that is distinct for normal fork progression compared to more persistent stalls, we sought to further investigate the helicase activity of MCM9 by mutating the Walker A site (K358A) and examining its effect on fork protection. Transfection of MCM9(K358A) into nontreated 9^KO cells did not rescue fork protection (Fig. 7b, compare plots 1 and 5) unlike that for MCM9(BRCv⁻) above (Fig. 7b, plot 3). Instead, transfection of MCM9(K358A) did rescue fork protection only in the presence of HU (Fig. 7b, compare plots 2 and 6), suggesting that recruitment of RAD51 by MCM9, through the BRCv motif, and not direct helicase activity on its own is necessary to stabilized persistently stalled forks, whereas the helicase activity is utilized for normal fork progression.

## Discussion

Mutations in MCM8 and MCM9 have been clearly linked with infertility and primary ovarian insufficiency[15,16] as well as predispositions to a variety of cancers[49,50]. The MCM8/9 complex has been primarily correlated with a role in DSB repair from damage induced by MMC, cis-PT, or IR contributing to HR[19,21,24,30], however, MCM8/9 has also been detected directly at replication forks[27,28]. This prompted us to investigate whether MCM8/9 also participates during active replication to either protect, promote, or process stalled replication forks. Our results are consistent with a fork progression/protection role for MCM8/9 that occurs during active replication in the absence of any exogenous damage, responding to transient impediments, as well as during more severe replisome stalling induced by HU.

We can now show that MCM8/9 normally aids in maintaining fork progression (Fig. 8a, b) and that their absence results in severe fork instability leading to NSD and DSBs induced by MUS81 (Fig. 8c). Previously, targeted depletion of the MCM2 subunit of the MCM2-7

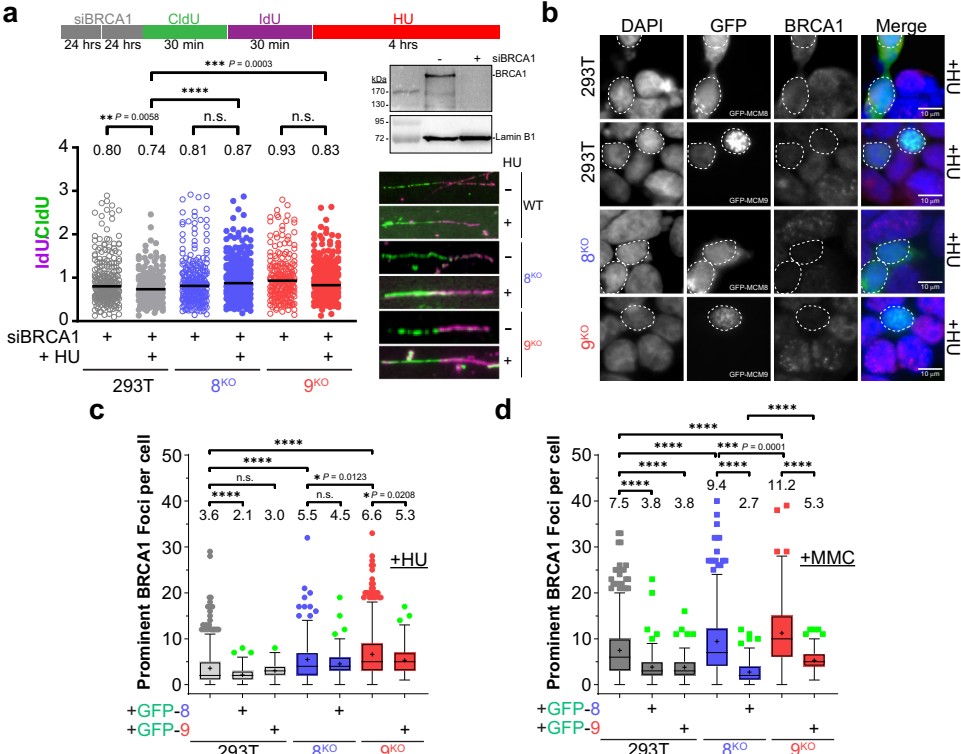

**Fig. 6 | BRCA1 functions with and is dependent on MCM8/9 to maintain replication fork integrity. a** 293T WT (gray), 8[KO] (blue), or 9[KO] (red) cells were transfected with siRNA twice for 24 h each to knockdown BRCA1 and verified by western blot. Cells were then sequentially incubated with CldU and IdU for the indicated time intervals (○) or followed by 2 mM HU for 4 h (●). DNA was spread by gravity to measure NSD by quantifying IdU and CldU lengths using Image J (>200 fibers). Representative fibers are shown to the right of the dot plots. The corresponding median ratios are reported at the top of the plots and by a black line embedded in the data. A Mann–Whitney two-sided U test was used to calculate *P* values (n.s. nonsignificant, *P < 0.05, **P < 0.01, ***P < 0.001, ****P < 0.0001). **b** GFP-MCM8 or GFP-MCM9 was transfected into each cell line and treated with 2 mM HU for 4 h (*n* = 2 independent experiments). Dashed white outlines of nuclei indicate cells successfully transfected and show a decrease in BRCA1 signal and foci. Larger

representative views of microscopy images of cells treated with HU or 3 μM MMC for 6 h are shown in Supplementary Fig. 11. **c, d** Prominent BRCA1 foci were quantified per cell for **c** HU (●) or **d** MMC (■) treated cells using Image J and plotted as a Tukey box and whiskers. Mean values are listed above the plots and with a + embedded in the data. Box boundaries represent the 1st and 3rd interquartile range (IQR); the line represents the median; the upper and lower whiskers are designated at a data point ≤ (±1.5× IQR); and individual data points outside the whiskers are shown. Only those cells transfected with GFP were used to count BRCA1 foci for HU (green, ●) or MMC (green, ■) treated conditions and plotted. The number of BRCA1 foci in more than 100 nuclei were quantified for each condition to calculate *P* values (n.s. nonsignificant, *P < 0.05, **P < 0.01, ***P < 0.001, ****P < 0.0001) by unpaired two-sided *t* test.

replication fork helicase complex resulted in continued replication and synthesis by MCM8/9, albeit at a significantly slower rate[27], consistent with our findings on promoting fork progression. MCM8 and MCM9 have been detected at higher abundances than even MCM2-7 (due to loading at dormant origins) or any other helicase at replication forks from coupled immunoprecipitation mass spectrometry studies[28]. Therefore, MCM8/9 facilitate replisome progression, and when they are absent, genome stability suffers as indicated by significantly more fork instability and γH2A.X foci and staining, even in the absence of any exogenous stressors. Although γH2A.X is commonly utilized as a marker for DSBs, it can also mark persistently blocked and reversed forks or single strand breaks directed by ATM[51]. In fact, our neutral comet analysis did not show significant DSBs in nontreated 8[KO] or 9[KO] cells, more consistent with single breaks or significant stalling/reversal indicated by γH2A.X. Combined, these results suggest that MCM8/9 is present and active within replisomes to aid in fork progression through challenging genomic stretches that may result in transient fork stalling/reversal processes (Fig. 8a, b).

Upon more severe fork stalling initiated by HU (or MMC), effects of MCM8 or MCM9 knockout on genome stability become more evident. MCM8 and MCM9 form nuclear foci when cells are treated with a variety of DNA damage agents (now including HU), and in their absence, more DSBs are detected by longer comet tail moments. To investigate the consequences to fork stability upon

knockout of MCM8 and MCM9, we utilized a suite of DNA fiber assays to specifically probe fork progression, reversal, protection, and resection. DNA fiber analysis shows that knockdown of SMAR-CAL1, and not HLTF, restores fork protection, overall implicating increased SMARCAL1 fork reversal activity when MCM8 or MCM9 are deficient, providing prevalent double-strand ends for γH2A.X binding. Resection of stalled forks is complicated by several nucleases acting with overlapping specificities and cooperativities to degrade a spectrum of reversed fork structures. Knockdown of MRE11 appears to have the greatest effect in restoring DNA fiber lengths in 8[KO] and 9[KO] cell lines, which was also corroborated with separate treatment with Mirin. However, siEXO1 also restored DNA fiber lengths, similar to that of siMRE11. siDNA2 was interesting in that in addition to restoring stability in 8[KO] and 9[KO] cells, it also completely restored the minimal sensitivity seen in WT cells. In our studies, the defects seen with HU-stalled forks in MCM8 or 9 deficient cells are linked to processes prior to fork resection and adding back either MCM8 or 9 did not rescue fork restart. This result is slightly different than that shown previously where MCM8/9 aided more directly in MRN resection processes of severely reversed forks[21]. Even so, in the absence of MCM8 or MCM9, stalled forks become extremely unstable, are actively reversed by SMARCAL1, and are then resected by a combination of coordinating nucleases to process all types of intermediates (Fig. 8c).

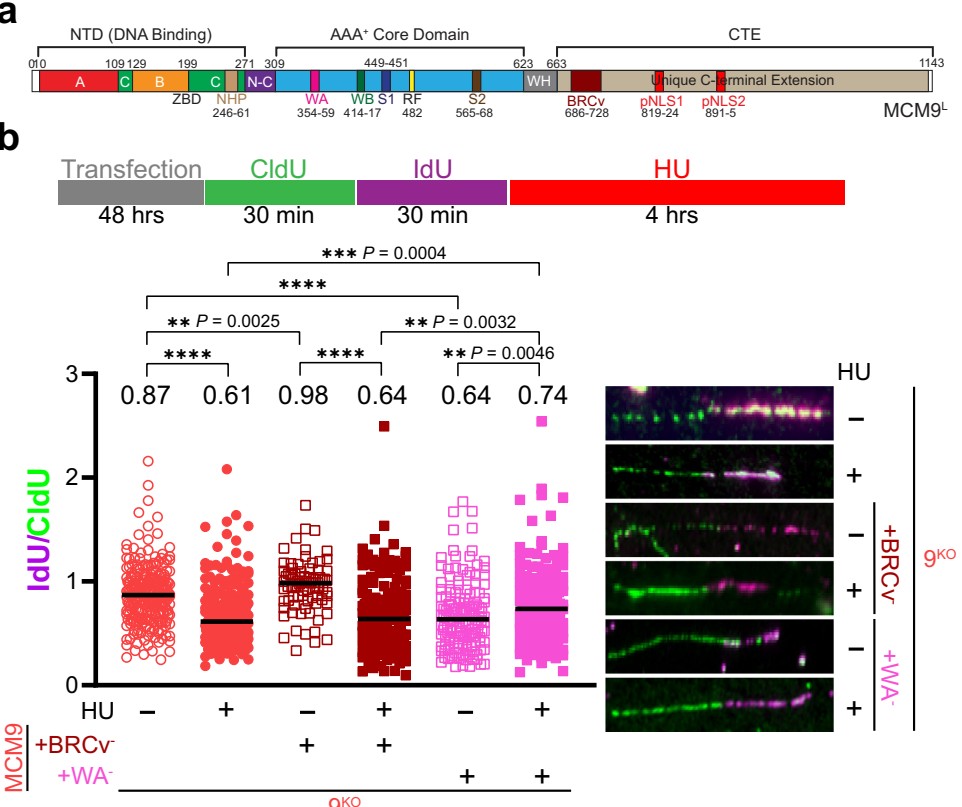

**Fig. 7 | The MCM9-BRCv motif is influential in stabilizing HU-stalled forks, but the ATPase activity is necessary for normal fork progression. a** Schematic of MCM9 showing domains and individual motifs. **b** 9$^{KO}$ cells (○) were transfected pEGFPC2-MCM9(FR687/8AA) (BRCv$^-$) (□, brown) or pEGFPC2-MCM9(K358A) (WA$^-$) (□, pink) for 48 h prior. Cells were then sequentially incubated with CldU and IdU for the indicated time intervals or followed by 2 mM HU for 4 h (● or ■). DNA was spread by gravity to measure NSD by quantifying IdU and CldU fiber lengths (>150 fibers) using Image J, and the corresponding median ratios are reported at the top of the plots and by a black line embedded in the data. Representative fibers are shown to the right of the dot plots. A Mann–Whitney two-sided U test was used to calculate *P* values (n.s. nonsignificant, **$P < 0.01$, ***$P < 0.001$, ****$P < 0.0001$).

As both MCM8$^{KO}$ and MCM9$^{KO}$ cell lines have increased γH2A.X foci in the absence and presence of exogenous agents, it is likely there is a spectrum of DNA intermediates with single strand gaps, stalled replisomes, and various reversed fork structures that require a MCM8/9 response. Once restart processes fail, those intermediates become targets of MUS81 cleavage (Fig. 8c). In fact, knockdown of MUS81 restored some fork protection under HU-stalled conditions, but not completely, highlighting again the competing roles of other nucleases and sub-pathways in this process. One of the hallmarks of MCM8 or MCM9 patient deficient cells was extreme sensitivities to MMC and the formation of broken, fused, and radial chromosomes[15,16], consistent with DNA end-joining processes occurring after more rampant fork cleavage by MUS81 and defects in HR.

Our evidence places MCM8/9 within or around the replisome actively responding to both transient and persistent stalling events to facilitate fork protection before significant processing can take place (Fig. 8a, b). Recruitment of RAD51 facilitated by the MCM9 BRCv motif, in particular, is influential in this dynamic protection process and likely stabilizes a subset of stalled forks that are not significantly reversed[31]. Even transfection of a catalytically inactive MCM9 stabilizes forks to a greater level than that of a BRCv$^-$ mutant during HU stalling, highlighting the importance of the BRCv motif to recruit RAD51 over that of the any associated ATPase activity utilized for fork progression. MCM8/9 antagonizes the effects of BRCA1 localization, which itself acts to stabilize stalled and reversed forks slightly further downstream to aid in their restart[52]. The results are consistent with a model whereby MCM8/9 utilizes its ATPase activity to promote fork progression through transient events, but during more persistent stalling events, MCM8/9 recruits RAD51/BRCA1/BRCA2 for fork protection (Fig. 8b).

Although BRCA1 and BRCA2 are generally assumed to play similar but temporal roles in fork protection, to sequentially recruit RAD51, emerging data suggests they are affected differently by MUS81[37,48]. While depletion of MUS81 confers fork protection in BRCA2$^{-/-}$ cells through a break-induced replication (BIR) pathway, it does not in BRCA1$^{-/-}$ deficient cells, highlighting a divergence in repair pathways, where BRCA1 can more adequately protect reversed forks from cleavage. Interestingly, more complete fork restoration required the elimination of both MCM8/9 and BRCA1, suggesting that fork reversal may not be possible in this situation as no suitable nuclease substrates are formed. In that case, an alternative pathway such as MUS81 cleavage and HR may be utilized. Therefore, MCM8/9 and BRCA1 appear to have non-redundant but mutually exclusive roles in maintaining fork stability, where MCM8/9 acts prior to BRCA1 recruitment but then leaves during the fork stabilization process.

Altogether, MCM8/9 mediates a pathway choice between fork progression and fork protection (Fig. 8b). The ATPase activity of MCM8/9 is utilized for normal fork progression within the replisome, possibly to restrict the formation of transiently reversed fork structures that can be recognized by stabilizer proteins. However, upon more severe stalling, the BRCv motif of MCM9 directs recruitment of RAD51 and BRCA1 to facilitate fork protection processes. In the future, it will be interesting to determine how MCM8/9 is incorporated within the replisome, better understand its DNA substrate specificity used

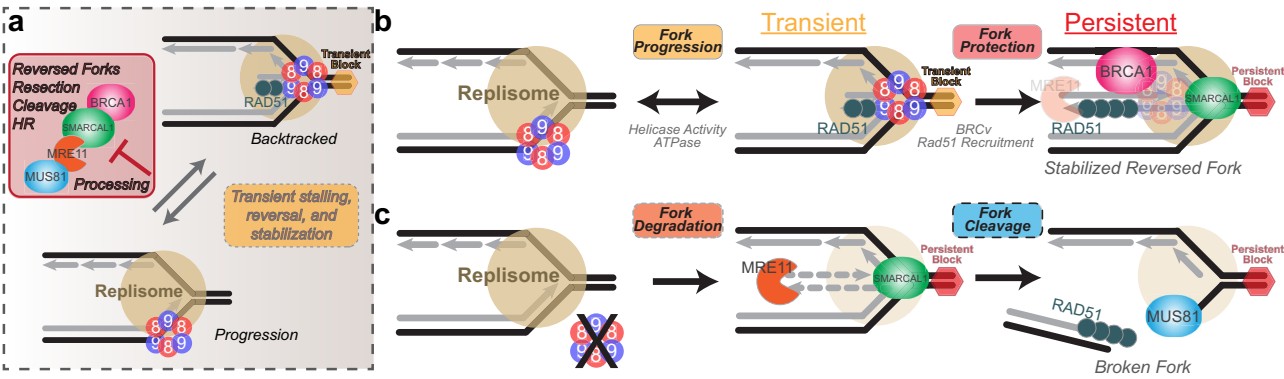

**Fig. 8 | A model for MCM8/9 aiding in normal fork progression or directing reversed fork protection during persistent obstacles. a** MCM8 and 9 participate within the replisome to overcome transient blocks (yellow hexagon) to fork progression and limit fork processing to maintain genomic integrity. **b** Should the replisome encounter persistent blocks (red hexagon), MCM8/9 works to recruit RAD51 and BRCA1 to stabilized and protect SMARCAL1 reversed and stalled forks from MRE11 (shaded) degradation. **c** In the absence of MCM8/9, any endogenous stressors allow for unregulated fork reversal and degradation through a SMARCAL1/MRE11/MUS81 pathway, resulting in increased fork cleavage and DSBs.

modulate reversed forks and interrogate the role of MCM9-recruited RAD51 in initiating downstream repair processes.

## Methods

### Cell culture

$8^{KO}$ and $9^{KO}$ in parental 293T cells were created using CRISPR/Cas9 technology and confirmed knockout by DNA sequencing and mitomycin C (MMC) sensitivity assays[31]. Two knockout clones for each were utilized for $8^{KO}$ (8B2 and 8B4) and $9^{KO}$ (9D6 and 9G10). Unless otherwise indicated, 8B2 and 9D6 were used primarily throughout. All cells were cultured at 37 °C with 5% $CO_2$ in Dulbecco's modified Eagle's medium (DMEM) (Corning Cellgro) supplemented with fetal bovine serum (FBS) (Atlanta Biologicals) at a 10% working concentration. Plasmid transfections (pEGFPc2-MCM8, pEGFPc2-MCM9, pEGFPc2-MCM9(FR687/8AA) were carried out using LPEI (ThermoFisher) as described for confocal microscopy[31]. The MCM8 or MCM9 gene was cloned into the pIRES2-EGFP using traditional restriction site cloning, *BglII/BamHI* and *XhoI/XmaI*, respectively. pEGFPc2-MCM9(K358A) was created by a modified QuikChange protocol and screened with a novel inserted restriction enzyme, *SmaI* (NENB). siRNAs were obtained from Dharmacon for siSMARCAL1 (5′-GCUUUGACCUUCUUAGCAAUU), siHLTF (5′-GGUGCUUUGGCCUAUAUCAUU), siBRCA1 (5′-CUAGAAAU-CUGUUGCUAUG), siMre11 (5′-ACAGGAGAAGAGAUCAACU), siDNA2 (5′-GUAACUUGUUUAUUAGACAUU), and siMus81 (5′-CAGCCCUG-GUGGAUCGAUAdTdT) or from Sigma for universal negative control #1 (SIC001). siRNAs and plasmids needed for reconstitutions assays of whole cell population assays were transfected using TransIT-X2 (Mirus) in Opti-MEM media following manufacturer's recommendation. Transfection efficiencies for GFP-containing plasmids were quantified using Countess II FL Automated Cell Counter (Invitrogen) equipped with EVOS light cubes for GFP (Supplemental Fig. 5). For more complete knockdown, cells were transfected twice for 24 h with the indicated siRNA before treatments. Cells were then treated with either 2 mM HU (Acros) for 4 h, 3 µM MMC (ThermoFisher) for 6 h, or 10 µM mirin (Sigma) for 6 h by adding agent directly to the media.

### Western blotting

Harvested cells were lysed in RIPA buffer (50 mM Tris pH 7.5, 150 mM NaCl, 0.1% SDS, 0.05% Triton, 10 mM DTT, 0.5 µM EDTA) and sonicated on ice. Protein content was quantified by BCA Assay (Boster Bio, AR01466) and stored at −20 °C. 30 µg of lysed protein was thawed on ice, electrophoresed on 8% or 10% acrylamide SDS-PAGE gel, and transferred onto PVDF or nitrocellulose in transfer buffer (25 mM Tris-HCl [pH 7.6], 192 mM glycine, 20% MeOH, 0.0375% SDS). The membrane was cut and blocked overnight in 5% powdered milk in 1xTBST at 4 °C, rocking. Following a wash with 1x TBST (used for all washes), membranes were incubated with their respective primary antibody [α-MCM8 (Proteintech, 16451-1-AP), α-MCM9 (ThermoFisher, PA5-113440), α-SMARCAL1 (Bethyl Laboratories, A301-616A), 1:100; α-HLTF (Bethyl Laboratories, A300-640A), 1:1000; α-BRCA1 (Santa Cruz, sc-6954), 1:250; α-MRE11 (Proteintech, 10744-1-AP), 1:500; α-DNA2 (Invitrogen, PA5-68167), 1:100; α-EXO1 (Bethyl Laboratories, A302-640A), 1:2000; α-MUS81 (Abcam, ab14387), 1:1000; α-GAPDH (Pierce, MA5-15738), 1:20,000, α-Lamin B1 (Proteintech, 12987-1-AP), 1:5000, α-β-actin (Abcam, ab82227), 1:10,000 for 2 h, rocking, at room temperature. The membranes were washed three times and incubated with secondary antibodies (goat anti-rabbit HRP Novex, A16096) (goat anti-mouse HRP, Novex, A16072), ranging from 1:1000–10,000, for 1 h, rocking, at room temperature. Three more washes were performed before addition of luminol reagents (Santa Cruz) and/or imaging with a Typhoon FLA9000 or ImageQuant LAS 4000 (Cytiva, Marlborough, MA) imager.

### DNA fibers

DNA fiber assays were performed as described previously with slight optimization modifications[42,53]. Briefly, cells were treated sequentially with 50 µM CldU (MP Biomedicals) and 500 µM IdU (TCI America) nucleotide analogs for indicated times, with a gentle wash with 1X PBS in between nucleotide incubations, prior to (unless indicated otherwise) treatment with DNA damaging or fork stalling agents. Cells were harvested after 2 washes with 1X PBS, pelleted, and stored at −20 °C before spreading. Cells were spread by gravity on silanized microscope slides by mixing 2 µL of cell suspension with an 8 µL drop of DNA fiber lysis buffer (200 mM Tris-Cl, pH 7.5; 50 mM EDTA; and 0.5% SDS). Drops were allowed to dehydrate for 10–20 min prior to spreading. Fiber spreads were then allowed to dry completely and were fixed to the slide by incubating in a 1:3 solution of methanol:acetic acid for 10 min before storage overnight at −20 °C. Fixed fibers were denatured for 25 min in 1 M NaOH solution followed by 2–3 washes in 1X PBS. Fibers were blocked for 30 min in fiber blocking buffer consisting of 1X PBS, 5% bovine serum albumin (BSA), and 0.1% Tween. Fibers were then incubated sequentially in humidified chambers with mouse (BD Bioscience, BD-347580) (1:50) and rat (Abcam, ab6325) (1:400) primary anti-BrdU antibodies in fiber blocking buffer for 1 h each with 2–3 washes in 1X PBS with 0.1% Tween between incubations. Fibers were simultaneously incubated in α-mouse-Cy3-conjugated (Abcam, 97035)

and α-rat 488-conjugated (Abcam, 150157) secondary antibodies (1:400) in fiber blocking buffer for 1 h. Slides were washed 2–3 times in 1X PBS with 0.1% Tween followed by mounting in mounting media consisting of 0.5X PBS, 25 mg/mL 1,4-Diazabicyclo[2.2.2]octane (DABCO), 1 mM ascorbic acid, and 90% glycerol. Mounted slides were sealed with clear polish. Fibers were then imaged on an Olympus IX-81 epifluorescence microscope with a 60X oil immersion objective and analyzed using Cell Sens Dimension 2 software. 100 or more fiber lengths were measured with ImageJ software (v1.52a, Rasband 1997–2016, 17 October 2015) to calculate IdU/CldU ratio values. For overall replication rate, slope values in $\mu$m m$^{-1}$ were converted to bp s$^{-1}$ using the known base pair distance (3.4 Å bp$^{-1}$) as the conversion factor. Scatter and violin plots were created using GraphPad Prism (v.9.4) and a Mann–Whitney U test was conducted to analyze statistical significances unless indicated otherwise.

## FACS analysis

293T, 8$^{KO}$, or 9$^{KO}$ cells were synchronized at the beginning of S-phase using a double thymidine block. Adherent cells were grown to 40% confluency in 10 cm$^2$ dishes with DMEM/10% FCS supplemented with 10 mM thymidine (TCI America) and cultured at 37 °C with 5% CO$_2$. After 18 h, the media was aspirated, and cells were washed three times with 10 mL pre-warmed PBS. Cells were released by the addition of unsupplemented DMEM/10% FCS for 8 h. Cells were again synchronized into G1/S phase by addition of DMEM 10%/FCS/10 mM thymidine. After 18 h, cells were washed 3x with pre-warmed PBS and released into fresh DMEM 10% FCS. Cells were harvested and fixed in 70% ethanol at indicated timepoints and stored at 4 °C. Cell pellets were stained using PI/RNase Staining Buffer (BD Biosciences, 550825) per manufacturers protocol. The cell cycle profile data was collected on a FACSVerse (BD Biosciences) using the propidium iodide channel. Cell cycle determination was analyzed using forward scatter (FSc) and side scatter (SSc), selecting for unaggregated live cells, graphed using FlowJo (BD Bioscience, v10), and presented using Adobe Illustrator (2021).

## Fluorescence and immunofluorescence imaging

Adherent cells on glass coverslips were washed in 1X PBS (2 times), fixed in 4% paraformaldehyde in PBS for 10 min, and permeabilized with 0.1% Triton X-100 in PBS (PBST) for 15 min. Cells were blocked overnight with 5% BSA in PBST at 4 °C. For immunofluorescence, coverslips were incubated with α-γH2A.X (Abcam, ab26350) (1:400) or α-BRCA1 (Santa Cruz, sc-6954) (1:50) dilution of primary antibodies in 2.5% BSA in PBST for 1 h at 37 °C. Cells were washed three times in PBST and incubated with 1:1000 dilution of the α-mouse Alexa647 (ThermoFisher, A-21235) secondary antibody followed and then washed three times with PBST. Cells were mounted in DAPI mountant (Prolong Gold, Thermo Fisher) and sealed with clear polish and imaged under a FV-1000 epifluorescence or FV-3000 confocal laser scanning microscope (Olympus Corp.). Images were processed with vendor included Fluoview (v.4.2b) or CellSens software (dimension 2). γH2A.X or BRCA1 foci from epifluorescence images were automatically counted from individually gated cells using identical thresholds that eliminated background noise using Image J, as described previously[31]. Foci per cell are presented in a box and whisker plot to identify the upper and lower quartiles, outliers, the median, and the mean. Data were analyzed for any statistically significant differences using a Mann–Whitney U test in GraphPad Prism unless otherwise indicated.

## Neutral comet assay

Comet assays were performed with the CometAssay® Electrophoresis System II (Trevigen, 4250-050-ES) following the manufacturer's protocol. Briefly, cells were harvested in 1X PBS. Cells were diluted in low-melting point agarose to a concentration of $1 \times 10^6$ cells/mL and 50 μL of cell solution was spotted on a microscope slide. Slides were placed

in the dark at 4 °C for 30–45 min to allow the agarose spot to dry. Slides were then immersed in Lysis Solution provided by the manufacturer for 30 min, then cooled to 4 °C for 60 min. Slides were then placed in 1X neutral electrophoresis buffer (50 mM Tris [pH 9.0], 150 mM sodium acetate) for 30 min. DNA was then electrophoresed at 21 V for 45 min and then the slides were immersed in DNA precipitation buffer (7.5 M Sodium Acetate and 95% ethanol) for 30 min at room temperature. Slides were then rinsed in water for 5 min followed by 70% ethanol for 5 min. Slides were then dried at 37 °C for 10–15 min followed by incubation in 1X SYBR Gold DNA stain (ThermoFisher, S11494) for 30 min at room temperature. Slides were briefly washed in water to remove excess stain and were allowed to dry completely at 37 °C. Slides were mounted with mounting solution as detailed above and imaged by epifluorescence microscopy. Percent DNA in the comet tails was measured with ImageJ software and tail moments were calculated according to Eqs. (1) and (2):

$$CTCF = IntDen - \left(Area_{cell} \times Fluor_{back}\right) \qquad (1)$$

$$Tail\ moment = \left(1 - \frac{CTCF_{head}}{CTCF_{whole}}\right) \times tail\ length \qquad (2)$$

where CTCF is the corrected total cell fluorescence for a comet head (head) or the whole comet (whole), IntDen is the integrated density, Area$_{cell}$ is the area of the selected cell, and Fluor$_{back}$ is the background mean fluorescence. Scatter plots were created using GraphPad Prism and a Mann–Whitney U test was conducted to analyze statistical significances unless indicated otherwise.

## Quantification and statistical analysis

Bars represent mean or median (as indicated) and the error bar represents the SEM of indicated numbers of independent experiments. Statistical analysis was performed by a Mann–Whitney U test in GraphPad Prism, as indicated. Scatter plots show all the individual data points; violin plots show the distribution of data with the first quartile, median, and the third quartile indicated; boxplots show the first quartile, median, third quartile, and whiskers which extend to 1.5× of the interquartile range with outliers shown.

## Reporting summary

Further information on research design is available in the Nature Research Reporting Summary linked to this article.

# Data availability

The data, plasmids, and cell lines generated in this study are available from the corresponding author upon reasonable request. Source data are provided with this paper.

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

## Acknowledgements

The authors wish to thank the Dr. Bernd Zechmann (Center for Microscopy and Imaging, Baylor University, Waco, Texas) and Dr. Michelle Nemec (Molecular Biosciences Center, Baylor University, Waco, Texas) for technical support during this work. We thank Elizabeth Jeffries for cloning the pIRES2-EGFP-MCM8 and pIRES2-EGFP-MCM9 plasmids. This work was supported by Baylor University and a NIH R15 (GM13791 to M.A.T.).

## Author contributions

W.C.G. performed and analyzed most of the DNA fiber assays, comet assays, and some of the confocal microscopy. D.R.M. provided the $8^{KO}$ and $9^{KO}$ cell lines, performed the FACS analysis, and western blots. K.N.K. did the siBRCA1 fiber assay, transfections, and western blots. R.B. performed transfections, comet assays, and western blots. A.E. performed some of the DNA fiber stability and replication rate assays. M.A.T. performed the γH2A.X and BRCA1 immunofluorescence assays and quantified the associated foci. M.A.T. and W.C.G. conceived the project and wrote the manuscript with input from all the authors. M.A.T. supervised the project.

## Competing interests

The authors declare no competing interests.

## Additional information

**Supplementary information** The online version contains

supplementary material available at https://doi.org/10.1038/s41467-022-32583-8.

