## [Peer Review File · Nature Communications]

A multi-functional role for the MCM8/9 helicase complex in maintaining fork integrity during replication stressEditorial Note: Parts of this Peer Review File have been redacted as indicated to maintain the confidentiality of unpublished data.

REVIEWER COMMENTS

Reviewer #1 (Remarks to the Author):

Although MCM8/9 associate with the replisome and replication forks stall more in the absence of MCM8 or 9, the role of MCM8/9 at the replication fork is unknown. This manuscript by Griffin et al describes the role of MCM8/9 at the replication fork. They show that replication slows and markers of DNA damage increase in the absence of MCM8 or 9. MCM8/9 prevent degradation of DNA at replication forks by MRE11, EXO1, and DNA2 after fork reversal by SMARCAL1 but not HLTf. MCM9 helicase activity is important for stabilizing replication forks under normal conditions, but with replication stress, the helicase activity is no longer necessary and the BRCv motif which recruits RAD51 becomes critical. In general, the data support the conclusions and would likely be of broad interest. However, there are some issues which I think should be addressed before publication.

1. For experiments where MCM8 or 9 is transfected into cells, a western blot showing expression levels would help to determine whether expression is similar to endogenous, overexpressed, or underexpressed. In Figure 2, tail moment is the same for untreated and HU treated WT cells, but in Figure S1, for 9KO cells with MCM9 expressed from a plasmid, the tail moment appears to be significantly increased with HU. This leads me to wonder whether differences in expression of MCM9 between the WT and transfected cells might explain the lack of recovery from expression of MCM9 in 9KO cells. Figure 9 is similar with addition of MCM8-GFP or MCM9-GFP to 8KO or 9KO cells not recovering fork restart activities. Is a different level of expression compared to the WT cells affecting fork restart? If not, are the GFP tags affecting the activity?

2. On p8, line 176. Reversal by SNF2 fork remodeling enzymes seems to be an overly broad conclusion since MCM8/9 appear to be involved in the same pathway as SMARCAL1 but not HLTf and both are SNF2 enzymes.

3. I am confused by the interpretations of the BRCA1 data. Why would cells lacking both BRCA1 and MCM8/9 may be unable to form reversed forks? Is there any evidence that BRCA1 or MCM8/9 is involved in fork reversal (either in this manuscript or published)? Figure 6b-d shows MCM8/9 reduce formation of BRCA1 foci. This data led to the conclusion that MCM8/9 antagonize recruitment of BRCA1 during normal replisome progression. However, this seems inconsistent with the data in Figure 4 and Figure 6a. BRCA1 and MCM8/9 both prevent DNA degradation at the replication fork. So if MCM8/9 antagonizes BRCA1 during normal replisome progress, why is there DNA degradation in Figure 4 in 8KO and 9KO cells in the untreated conditions? If MCM8/9 antagonize BRCA1, the DNA at the replication fork would be more likely to be degraded when MCM8/9 are present than when they are absent, but that is not the result in Figure 4. Similarly, for the data in Figure 6a, since BRCA1 and MCM 8/9 are all protective against DNA degradation, why would loss of both MCM8 or 9 and BRCA1 result in protection of the DNA from degradation?

4. In Figure 8, the untreated 8KO and 9KO cells show some degradation with Mirin that is absent with siMRE11. Is this due to Mirin only inhibiting MRE11 exonuclease activity? If so, are both the MRE11 exo and endonuclease activity involved in degradation?

5. It is not clear to me how DNA2 could be a backup nuclease that processes a specific subset of forks since knockdown of DNA2 prevented degradation of the replication fork under all conditions tested.

6. The statistical analysis is well described in the figure legends, and the methods state that at least 100 fibers are quantified for each condition. Adding the number of replicates for each experiment to the figure legends (or methods if it's uniform throughout) would help to determine the rigor or experiments.

7. The model is difficult to understand as drawn. In a, my initial interpretation was that backtracking inhibited reversed forks, resection, cleavage, and HR. I think the intent is for MCM8/9 to be inhibiting RF, resection, cleavage and HR, although I'm not sure. I did not figure out what the yellow hexagon was until I proceeded to panel b. Also in b, I'm not sure why MCM8/9 and MRE11 are shadows in the last panel.

Reviewer #2 (Remarks to the Author):

The MCM8/9 helicase is a paralog complex of the replicative MCM2–7 helicase and is involved in homologous recombination (HR) in the somatic and germline cells. MCM8/9 functions in HR after the breakage of DNA replication forks in the somatic cells. In germline cells, it works in meiotic recombination, promoting crossover, and loss of MCM8/9 results in infertility. In the manuscript by Griffins et al., the authors propose that MCM8/9 is also involved in protecting nascent strand degradation (NSD) at replication forks.

The authors generated knockout 293T cell lines (MCM8KO and 9KO) and investigated fork speed, S phase progression, and accumulation of DNA damage (Figures 1-3). After confirming that MCM8KO and 9KO accumulated more DNA damage, they investigated NSD using DNA fiber assay. In the MCM8KO and 9KO cells, NSD was enhanced, and this was more evident when treated with HU (Figure 4). They showed MCM8/9 protected NSD caused by SMARCA1 and BRCA1, but not HLTF (Figures 5 and 6), and the BRCv domain within MCM9 was required to protect when persistent fork reversal was induced by HU (Figure 7). In addition, MCM8/9 prevented nascent strands from Mre11, Exo1, and Dna2, which degrade the nascent strand at stalled replication forks (Figure 8) and protected from Mus81, which cleaves stalled replication forks in HU (Figure 9).

It is an intriguing possibility that MCM8/9 is involved in the protection of NSD at replication forks. However, the proposed model is not strongly supported because of methodological and technical problems. I will summarize the key issues below.

The MCM8KO and 9KO lines

The authors used only one KO line each for MCM8 and 9. There is a possibility that the author might have picked a clone having other mutations. Therefore, it is essential to use at least two independent KO clones for key experiments (such as Figures 1 and 4). I understand the KO lines were initially described in their previous paper (McKinzev et al. JCB, 2021). However, it was still unclear how the KO lines were generated. Which exon was targeted by CRISPR/Cas9? Genotype? Immunoblot data? There is the histone-chaperon ASF1A gene located within an intron of MCM9. Was ASF1A affected by the loss of MCM9?

Rescue experiments

To characterize KO cell lines, it is important to carry out rescue experiments. In Figures 1-3, such rescue experiments are missing. The authors have done rescue experiments in Figures 4 and 7. However, it is unclear how the assays were done. I understand that pEGFPc-MCM8/9 was transfected. But judging from the data shown in Figure 2a, transfection efficiency was not high. Because they needed to use a cell population for DNA fiber assay, it was impossible to use the cells soon after transfection. Did the authors isolate a clone or select transfected cells by drug selection? How was the expression level of the transgene compared to the endogenous gene?

Calculation of fork speed

Because the authors used only CldU (but not double labelling), it is impossible to calculate fork speed from the data shown in Figure 1b-e because new origin firing occurred during the experiments. The

track length shows the sum of fork progression and new origin firing. This is supported by the fact that the track length increased abruptly at later time points (Figure 1b, 60 and 75 min).

Providing appropriate control in the same panel

The authors provided NSD data of MCM8KO and 9KO in Figure 4a, and similar NSD data of the rescue cell lines in Figure 4b. Looking at these data carefully, NSD in MCM8KO was not rescued, somewhat worse, by introducing the MCM8 transgene (plot 3 in Figure 4a vs plot 1 in figure 4b). The provided data was not convincing to conclude that the NSD phenotype in MCM8KO was rescued by introducing MCM8 transgene. Overall, it was difficult for me to understand each figure because similar data were segmented into different panels and figures. In some cases, they did not contain an appropriate control.

Other issues

1. Recent reports showed that MCM8/9 works with another protein, HROB/MCM8IP, in HR (Hustedt et al. G&D, 2019; Huang et al. Nat. Commun., 2020). This point was not mentioned at all in the present manuscript. It is also interesting to clarify if HROB/MCM8IP is also involved in NSD.
2. Immunofluorescent pictures shown in Figures 2, 3, and 6 are too small and unclear. It was difficult to see cells and nuclear foci.

Reviewer #3 (Remarks to the Author):

In this work, Griffin and colleagues analyzed whether loss of MCM8 and 9, two MCM-like factors involved in recombination and replication, affects replication.

By using single-molecule replication assays (mostly) and analysis of DNA damage, they find that MCM8/9 affects replication fork progression in untreated cells and impair fork stability on HU. They also provide data supporting that fork degradation occurs downstream SMARCA1-mediated fork reversal and is dependent on BRCA1. Strikingly, they show that the ability of MCM9 to bind RAD51 but not its helicase activity is important for fork stability on HU.

The mechanisms controlling stability of perturbed forks are crucial for integrity of the genome and also involved in chemosensitivity of cancer cells. Thus, the more details we have on the pathways/factors involved better we will understand how genome stability is maintained.

That said, the work of Griffin and colleagues, although potentially interesting to the field, falls short in providing strong data supporting the hypothesis and also in identifying how MCM8/9 would ensure fork integrity.

More experiments are needed to better outline how MCM8/9 protects forks and to define if the only effect on replication fork progression is related to fork instability or rather to reduced recombination away from the fork since MCM8/9 is involved in recruiting RAD51.

The manuscript also would benefit from extensive revision in the structure and in data presentation.

Major comments:

In some cases, a better understanding of the message would benefit from a change in the order of the results and figures (e.g. Fig8).

Figure 1. Fiber experiments have been performed in untreated conditions. Data show a clear reduction in the replication rates when MCM8 and MCM9 are knocked out however just single clones and a single cell line have been used. I would recommend authors to perform additional add-back experiments to formally demonstrate that the phenotype specifically relates to their KO. Moreover, while the single labeling strategy and multiple sampling time may be sufficient to obtain data on replication speed, I advise authors to also perform experiments using a standard dual-labeling approach, which might help them in identifying asymmetric forks that are a readout of fork stalling. The use of a dual labeling

strategy might also be of help in the understanding of why loss of MCM8 or 9 does not affect replication progression at short labeling time while reducing greatly tract length with time. This is an interesting point that the authors should clarify.

Figure 3. In Fig2b authors do not show any difference in the level of DSBs between WT and KO cells in untreated conditions while in Fig3a-c they observe more H2AX+ cells but they state that the H2AX result is consistent with reduced fork progression. I suggest to specify the differences in the conclusion at the end of the paragraph.

Figure 4. Authors state that, in untreated conditions and on HU, the IdU/CldU ratio in KO cells is reduced with respect to the WT and that may be related to fork instability, however, the relevant controls (i.e. nuclease KD or inhibition) are shown in Fig.8. They should be shown in Fig4 instead. Moreover, it would be useful to have an "internal" control, that is a condition known to induce fork degradation, to determine how strong is the phenotype conferred by MCM8/9 loss and if it is epistatic with known conditions leading to fork degradation (beside the experiments correlating the MCM8/9's role with that of BRCA1 reported in Fig6).

Figure 5. Authors show that depletion of SMARCAL1 but not HTLF prevents signs of fork degradation in KO cells. First of all, I would recommend authors to also show data from mock-depleted points alongside data from depleted ones because this makes understanding of the data easier to readers. That said, it is needed to use an additional way to downregulate fork reversal to substantiate their claim.

Figure 6. Rationale of the experiments shown in Fig6 is completely unclear to me. Why do you expect BRCA1 loss mitigates the MCM8 or 8 KO? At least to me, It sounds like the hypothesis is that BRCA1 and MCM8/9 are epistatic or not. Alternatively, as data suggest, if loss of BRCA1 restores the WT phenotype, authors should provide experiments supporting a mechanism. For instance, does BRCA1 depletion prevent loading of exonucleases at deprotected forks in the absence of MCM8/9? How do the authors reconcile the role of BRCA1 in protecting forks with the apparent reduction in BRCA1 foci in response to HU when MCM8/9 are KO?

This is a crucial point and the presented data are not fully supportive of the proposed difference between unstressed vs. stressed.

As suggested above, it absolutely needed to show mock-depleted data in the graphs to avoid continuous back and forth through figures.

Figure 7. Authors performed experiments to determine if the BRCv motif or the helicase activity of MCM9 was involved in fork stabilization. This is an important point that would require previous analysis of whether MCM9 (or MCM8) destabilizes RAD51 filaments at the reversed fork.

Figure 8. This figure should be merged with Fig4. Authors should include mock-depleted data in the graphs. Authors show that basically each exonuclease is apparently involved in degrading reversed forks in KO cells. Usually, destabilized forks get degraded by MRE11-EXO1 while DNA2 is mostly involved in the limited resection thought to take place in WT setting. How do authors reconcile with this? Little explanation is provided in discussion.

Minor comments:

A general suggestion is to revise wording and structure of some sentences in the text and in the legends. I found some redundancy here and there.

- I think the introduction might be improved by avoiding some redundancies existing when talking about previous evidence of an MCM8/9 role during replication
- I found few typos throughout the manuscript (e.g. line 123 "genomic instability" where it should

read "genomic stability"

- Some figures can be merged also to make catching the result easier (e.g. Fig2 + Fig3; Fig4+Fig5) or moved as supplemental information (e.g. Fig.2a can be put in the supplemental material)
- Line 149. The first sentence of the paragraph is an overstatement. The previous paragraph just shows that KO cells replicate slower when unchallenged and have or not have depending on the assay more DNA damage.
- Titles of some figures are not consistent with the key finding.

Response to Reviewers

Reviewer #1 (Remarks to the Author):

Although MCM8/9 associate with the replisome and replication forks stall more in the absence of MCM8 or 9, the role of MCM8/9 at the replication fork is unknown. This manuscript by Griffin et al describes the role of MCM8/9 at the replication fork. They show that replication slows and markers of DNA damage increase in the absence of MCM8 or 9. MCM8/9 prevent degradation of DNA at replication forks by MRE11, EXO1, and DNA2 after fork reversal by SMARCAL1 but not HLTF. MCM9 helicase activity is important for stabilizing replication forks under normal conditions, but with replication stress, the helicase activity is no longer necessary and the BRCv motif which recruits RAD51 becomes critical. In general, the data support the conclusions and would likely be of broad interest. However, there are some issues which I think should be addressed before publication.

1. For experiments where MCM8 or 9 is transfected into cells, a western blot showing expression levels would help to determine whether expression is similar to endogenous, overexpressed, or underexpressed. In Figure 2, tail moment is the same for untreated and HU treated WT cells, but in Figure S1, for 9KO cells with MCM9 expressed from a plasmid, the tail moment appears to be significantly increased with HU. This leads me to wonder whether differences in expression of MCM9 between the WT and transfected cells might explain the lack of recovery from expression of MCM9 in 9KO cells. Figure 9 is similar with addition of MCM8-GFP or MCM9-GFP to 8KO or 9KO cells not recovering fork restart activities. Is a different level of expression compared to the WT cells affecting fork restart? If not, are the GFP tags affecting the activity?

The endogenous levels of MCM8 and MCM9 are very low (see Jeffries 2013 Gene), and we cannot consistently detect native MCM8 or MCM9 using our custom (and purified) antibodies. Therefore, any transfections would be overexpressed from native conditions, especially using the CMV promoter in these plasmids, which is commonplace for most other replacement assays. Here is a western blot (for review only that shows overexpression of GFP-MCM8 or GFP-MCM9 in their respective knockout cell lines.

[redacted]

It is possible that differences in expression can account for differences in absolute values for tail moments (or other assay differences), however the trends for the rescue experiments are

representative and clear. In Supp. Fig. 5, adding back MCM9 (on an IRES plasmid, so no tag) (and now also with adding back MCM8) does reduce (*i.e.* rescue) the tail moment length. Similar, rescue experiments are also shown that increase DNA fiber ratios (*i.e.* stability) in Fig 4b (plots 6&10) and for rescue of replication rates in Supp. Fig 2. It would be really difficult to titrate MCM8 or 9 levels that low as to be representative of endogenous. Both transfection efficiency (>80%) and altered expression levels could account for incomplete rescue.

2. On p8, line 176. Reversal by SNF2 fork remodeling enzymes seems to be an overly broad conclusion since MCM8/9 appear to be involved in the same pathway as SMARCAL1 but not HLTF and both are SNF2 enzymes.

OK, this may be too broad. We have changed this to “These data suggest that, in the absence of MCM8/9, replication fork stability is reduced and prone to degradation following more prevalent fork reversal.”

3. I am confused by the interpretations of the BRCA1 data. Why would cells lacking both BRCA1 and MCM8/9 may be unable to form reversed forks? Is there any evidence that BRCA1 or MCM8/9 is involved in fork reversal (either in this manuscript or published)? Figure 6b-d shows MCM8/9 reduce formation of BRCA1 foci. This data led to the conclusion that MCM8/9 antagonize recruitment of BRCA1 during normal replisome progression. However, this seems inconsistent with the data in Figure 4 and Figure 6a. BRCA1 and MCM8/9 both prevent DNA degradation at the replication fork. So if MCM8/9 antagonizes BRCA1 during normal replisome progress, why is there DNA degradation in Figure 4 in 8KO and 9KO cells in the untreated conditions? If MCM8/9 antagonize BRCA1, the DNA at the replication fork would be more likely to be degraded when MCM8/9 are present than when they are absent, but that is not the result in Figure 4. Similarly, for the data in Figure 6a, since BRCA1 and MCM 8/9 are all protective against DNA degradation, why would loss of both MCM8 or 9 and BRCA1 result in protection of the DNA from degradation?

Reviewer 3 also had a similar question, so please see further discussion there.

There is no evidence (to our knowledge) that BRCA1 or MCM8/9 can reverse forks directly. However, In the absence of SMARCAL1 and/or HLTF, forks can still be remodeled by Rad51 to resemble a subset of reverse forks (Zellweger 2015), and this can be done in the absence of the Rad51 mediator, BRCA2 (Kolinjivadi 2017). However, BRCA2 is required for stabilizing Rad51 bound to reversed forks in a protection role (Schlachter Cell 2011). BRCA1 also acts to stabilize replication forks but is much less mechanistically characterized (Schlachter Cancer Cell 2012). We are proposing a synergistic counter relationship for MCM8/9 and Rad51/BRCA1, such that when MCM8/9 are present, they are generally promoting fork progression (Fig. 1), unless severe stress is encountered and then MCM8/9 can facilitate a controlled hand off of the fork to Rad51/BRCA2/BRCA1 to begin fork reversal and protection. When MCM8/9 are absent, SMARCAL1 directs more uncontrolled fork reversal that end up being less protected and therefore unstable and mimics a DSB end, b/c Rad51 and BRCA1 are not recruited efficiently. MCM8/9 and BRCA1 are protective in different ways. The presence of MCM8/9 reduces the frequency of reversed forks, but when reversal is the only option, MCM8/9 directs Rad51 in fork remodeling and BRCA1 stabilizes reversed forks from degradation. In Figure 4b, forks are unstable because in the absence of MCM8/9, there is more fork reversal that is not carefully handed off to Rad51/BRCA2/BRCA1 and therefore unprotected. But in Figure 6a, the absence of MCM8/9 and siRNA knockdown of BRCA1 does not allow for fork reversal and provides

limited reversed fork template for various nucleases. Therefore, we suspect that BRCA1 is implicated in stimulating controlled fork reversal as handed off from MCM8/9.

We agree this is a complex relationship that we are beginning to parse out, but it is also novel with regards to MCM8/9 regulating the fork progression-reversal process depending on the severity of the block. We have attempted to make this clearer in the revised manuscript. Please see updated in the Results surrounding Fig 4 and 6 and then in the Discussion.

4. In Figure 8, the untreated 8KO and 9KO cells show some degradation with Mirin that is absent with siMRE11. Is this due to Mirin only inhibiting MRE11 exonuclease activity? If so, are both the MRE11 exo and endonuclease activity involved in degradation?

Yes, this is a good point, it could also be MRE11 endonuclease, especially if forks are less protected. In the text (originally lines 253-55), we described this as other nuclease such as EXO1, CtIP, and DNA2 being active. And in the subsequent Fig 4 (old Fig 8) panels, we show that both EXO1 and Dna2 have a role in degradation. Likely, there are various undescribed reversed fork substrate specificities for each of the nucleases that are being combined in this experiment. We have updated this statement to include the possibility of more endonuclease cleavage events with Mirin over that or siMRE11.

5. It is not clear to me how DNA2 could be a backup nuclease that processes a specific subset of forks since knockdown of DNA2 prevented degradation of the replication fork under all conditions tested.

We did not explain this specific enough in the original submission. We have added several clarifying statements and additional references discussing the complexity of the nuclease issue, but for DNA2 in these studies, we have updated the text to say “implicating DNA2 as an additional nuclease that can process or degrade replication forks through multiple mechanisms.” And then go on to describe the possibilities. However, the main point remains, that MCM8/9 acts to prevent multifaceted nucleolytic degradation by stimulating fork protection.

6. The statistical analysis is well described in the figure legends, and the methods state that at least 100 fibers are quantified for each condition. Adding the number of replicates for each experiment to the figure legends (or methods if it's uniform throughout) would help to determine the rigor of experiments.

All fiber assays were analyzed from more than 100 and typically 150-200 fibers. Some fiber assays were repeated more than once for validation; however, others were performed simultaneously with WT populations as the control. In response to Reviewer 2&3, we have repeated several conditions of these fiber assays with not only the primary clones (Supp Fig 2, Supp Fig 7) used initially, but now also with second MCM8 or 9 knockout clones (Supp Fig 1, Supp Fig 7) to show reproducibility and consistency. We have added more details of the replicates, conditions, and statistical parameters in all the Figure legends where absent.

7. The model is difficult to understand as drawn. In a, my initial interpretation was that backtracking inhibited reversed forks, resection, cleavage, and HR. I think the intent is for MCM8/9 to be inhibiting RF, resection, cleavage and HR, although I'm not sure. I did not figure out what the yellow hexagon was until I proceeded to panel b. Also in b, I'm not sure why MCM8/9 and MRE11 are shadows in the last panel.

Yes, your second thought is what we were trying to convey. The presence of 8/9 is there to prevent reversed forks from forming during normal fork progression. We have updated the figure and the legend to better explain the hexagons as a transient or persistent stalling event and also explain that during persistent stalls, MCM8/9's role is reduced and hands off the fork to Rad51/BRCA1/BRCA2 for stabilization of reversed forks and prevent MRE11 degradation. In the absence of MCM8/9, there is not control or regulation of fork reversal and aberrant degradation occurs leading to cleavage and DSBs. We have also added boxes for "Fork Progression", "Fork Protection", "Fork Degradation", and "Fork Cleavage" to better direct the reader.

Reviewer #2 (Remarks to the Author):

The MCM8/9 helicase is a paralog complex of the replicative MCM2–7 helicase and is involved in homologous recombination (HR) in the somatic and germline cells. MCM8/9 functions in HR after the breakage of DNA replication forks in the somatic cells. In germline cells, it works in meiotic recombination, promoting crossover, and loss of MCM8/9 results in infertility. In the manuscript by Griffins et al., the authors propose that MCM8/9 is also involved in protecting nascent strand degradation (NSD) at replication forks.

The authors generated knockout 293T cell lines (MCM8KO and 9KO) and investigated fork speed, S phase progression, and accumulation of DNA damage (Figures 1-3). After confirming that MCM8KO and 9KO accumulated more DNA damage, they investigated NSD using DNA fiber assay. In the MCM8KO and 9KO cells, NSD was enhanced, and this was more evident when treated with HU (Figure 4). They showed MCM8/9 protected NSD caused by SMARCA1 and BRCA1, but not HLTF (Figures 5 and 6), and the BRCv domain within MCM9 was required to protect when persistent fork reversal was induced by HU (Figure 7). In addition, MCM8/9 prevented nascent strands from Mre11, Exo1, and Dna2, which degrade the nascent strand at stalled replication forks (Figure 8) and protected from Mus81, which cleaves stalled replication forks in HU (Figure 9).

It is an intriguing possibility that MCM8/9 is involved in the protection of NSD at replication forks. However, the proposed model is not strongly supported because of methodological and technical problems. I will summarize the key issues below.

The MCM8KO and 9KO lines

The authors used only one KO line each for MCM8 and 9. There is a possibility that the author might have picked a clone having other mutations. Therefore, it is essential to use at least two independent KO clones for key experiments (such as Figures 1 and 4). I understand the KO lines were initially described in their previous paper (McKinze et al. JCB, 2021). However, it was still unclear how the KO lines were generated. Which exon was targeted by CRISPR/Cas9? Genotype? Immunoblot data? There is the histone-chaperon ASF1A gene located within an intron of MCM9. Was ASF1A affected by the loss of MCM9?

We have characterized and validated these KO cell lines in McKinze JBC 2021. In that paper (Fig. S4), we show both the genotyping sequencing and functional assays showing sensitivities to MMC. Exon 1 was targeted for MCM9, and Exon 3 was targeted for MCM8. ASF1A is in the seventh intron of MCM9 (and in an anticoding orientation) and is well away from the CRISPR target locus.

That said, we have now added confirmatory experiments with second KO clones of MCM8 and MCM9, as requested. New Supp. Fig. 1 shows a similar deficiency in replication rate for these two other clones compared with Fig 1a-b. Furthermore, we have used these secondary clones to examine fork stability +/-HU in a new Supp Fig 7, which shows similar fork instability to that of Fig. 4a-b.

Rescue experiments

To characterize KO cell lines, it is important to carry out rescue experiments. In Figures 1-3, such rescue experiments are missing. The authors have done rescue experiments in Figures 4 and 7. However, it is unclear how the assays were done. I understand that pEGFPc-MCM8/9 was transfected. But judging from the data shown in Figure 2a, transfection efficiency was not high. Because they needed to use a cell population for DNA fiber assay, it was impossible to use the cells soon after transfection. Did the authors isolate a clone or select transfected cells by drug selection? How was the expression level of the transgene compared to the endogenous gene?

Related to Figure 2, rescue experiments for Comets were performed in original Supplemental Figure 1 (now Supp. Fig. 5). Adding back MCM9 on an IRES plasmid reduced the Comet tail moment compared to 9KO cells alone suggesting a rescue of activity. We have now added the transfection back of pIRES2-MCM8 to the 8KO cell line and can show a similar rescue in these Comet assays (updated Supp. Fig 5). Moreover, we have also added additional add back experiments to show a rescue of the DNA replication rate (New Supp. Fig. 2). We had also shown in the original submission the rescue of fork stability in DNA fiber assays when adding back MCM8 or MCM9 to respective knockout cell lines (updated Fig. 4B, plots 6&10). Finally, we have also added data for second knockout clones for MCM8 or MCM9 (Supp. Fig. 1 and Supp. Fig. 7) that show similar fork instabilities as the original clones (to Fig. 4b) in this manuscript.

Altogether, we feel that the KO cell lines are validated from 1) a previous publication showing sequencing data and phenotypic characterization (McKinze et al 2021) 2) near identical phenotypic results from second clonal knockout populations, and 3) from successful rescue experiments in COMET and fiber assays.

We utilized two different transfection approaches (now detailed better in the M&M). For any DNA fiber or restoration assays, we used the commercial transfection reagent TransIT-X2 (Mirus) which gives ~80% transfection efficiency (New Supp. Fig. 3a-b). For confocal microscopy experiments examining foci or immunofluorescence, we used traditional PEI transfection which gives 30-40% efficiency. In that case, PEI transfection was useful in having cells in the same culture that were untransfected for direct comparisons of BRCA1 foci/fluorescence (+/-GFP) (Fig. 6b-d).

Transfection back into KO cells with MCM8 or 9 constructs will definitely overexpress (CMV promoter) MCM8 or MCM8 compared to endogenous levels. (See also response to Reviewer 1, comment 1).

Calculation of fork speed

Because the authors used only CldU (but not double labelling), it is impossible to calculate fork speed from the data shown in Figure 1b-e because new origin firing occurred during the

experiments. The track length shows the sum of fork progression and new origin firing. This is supported by the fact that the track length increased abruptly at later time points (Figure 1b, 60 and 75 min).

The reviewer is correct that when using a single labeling strategy, it is impossible to calculate fork speeds directly, however, it is possible to calculate the overall amount and rate of replication. Of course, this would include both fork speed directly, but also the activation of other bidirectional forks, which does complicate things. The text and Figure 1 were reworded to account for this caveat. In addition, we have now added a new Supplemental Figure 1 that shows similar replication rate deficiencies in another KO clone of MCM8 and MCM9 to show reproducibility and validation.

We have also added another Supplemental Figure 2 that includes an experiment that restores overall replication rates when adding back MCM8 or MCM9 to their respective knockout cell lines. A still open question is whether the absence of MCM8 or 9 allows for the activation of more origins, and we would prefer to reserve that result for another future manuscript. That said, all further DNA fiber assays are of the two-color variety that allow for an appropriate control pulse length.

Please see also response to Reviewer 3, 1st comment.

Providing appropriate control in the same panel

The authors provided NSD data of MCM8KO and 9KO in Figure 4a, and similar NSD data of the rescue cell lines in Figure 4b. Looking at these data carefully, NSD in MCM8KO was not rescued, somewhat worse, by introducing the MCM8 transgene (plot 3 in Figure 4a vs plot 1 in figure 4b). The provided data was not convincing to conclude that the NSD phenotype in MCM8KO was rescued by introducing MCM8 transgene. Overall, it was difficult for me to understand each figure because similar data were segmented into different panels and figures. In some cases, they did not contain an appropriate control.

While this is true that NSD was not rescued in the 8KO with transfection, equivalent rescue was also only just significant in the 9KO cells. The more influential result still ends up being the HU treated and rescued cells, where there is more obvious rescue in both cases.

Reviewer 3 had the same concerns regarding the organization of the plots and data, and so, we have substantially reorganized the Figures to better include appropriate controls within the same figure and still maintain some overall readability without overwhelming the reader (we hope).

- Old Fig 4A&B has been arranged to have all the fibers in 1 plot (New Fig 4b)
- Old Fig 8a-d (nucleases) has been incorporated into new Fig. 4c-f
- Old Fig 5(reversal) and Fig 8(restart) has been combined into a new Fig 5

Other issues

1. Recent reports showed that MCM8/9 works with another protein, HROB/MCM8IP, in HR (Hustedt et al. G&D, 2019; Huang et al. Nat. Commun., 2020). This point was not mentioned at all in the present manuscript. It is also interesting to clarify if HROB/MCM8IP is also involved in NSD.

We are also very interested in HROB/MCM8IP and their role at the replication fork. This manuscript was meant to first highlight the role of MCM8/9 in fork progression and then follow up with a subsequent manuscript on the relationship between MCM8/9 and HROB/MCM8IP using similar techniques focused on origin firing, fork progression $\leftarrow \rightarrow$ fork reversal/stabilization relationships.

2. Immunofluorescent pictures shown in Figures 2, 3, and 6 are too small and unclear. It was difficult to see cells and nuclear foci.

We made these images fairly large within the figure at the outset and we also colored them grey scale for better contrast. However, we have now zoomed in on select cells and updated Figure 2a, 3a-b, and 6b. The original images are now included as Supplemental Figs S4, S6, and S8 to show a larger population of cells..

Reviewer #3 (Remarks to the Author):

In this work, Griffin and colleagues analyzed whether loss of MCM8 and 9, two MCM-like factors involved in recombination and replication, affects replication.

By using single-molecule replication assays (mostly) and analysis of DNA damage, they find that MCM8/9 affects replication fork progression in untreated cells and impair fork stability on HU. They also provide data supporting that fork degradation occurs downstream SMARCAL1-mediated fork reversal and is dependent on BRCA1. Strikingly, they show that the ability of MCM9 to bind RAD51 but not its helicase activity is important for fork stability on HU.

The mechanisms controlling stability of perturbed forks are crucial for integrity of the genome and also involved in chemosensitivity of cancer cells. Thus, the more details we have on the pathways/factors involved better we will understand how genome stability is maintained.

That said, the work of Griffin and colleagues, although potentially interesting to the field, falls short in providing strong data supporting the hypothesis and also in identifying how MCM8/9 would ensure fork integrity.

More experiments are needed to better outline how MCM8/9 protects forks and to define if the only effect on replication fork progression is related to fork instability or rather to reduced recombination away from the fork since MCM8/9 is involved in recruiting RAD51.

The manuscript also would benefit from extensive revision in the structure and in data presentation.

Major comments:

In some cases, a better understanding of the message would benefit from a change in the order of the results and figures (e.g. Fig8).

Based on this comment and others below as well as from Reviewer 2, we have rearranged the figures as suggested.

- Old Fig 4A&B has been arranged to have all the fibers in 1 plot (New Fig 4b)
- Old Fig 8a-d (nucleases) has been incorporated into new Fig. 4c-f
- Old Fig 5(reversal) and Fig 8(restart) has been combined into a new Fig 5

Figure 1. Fiber experiments have been performed in untreated conditions. Data show a clear reduction in the replication rates when MCM8 and MCM9 are knocked out however just single clones and a single cell line have been used. I would recommend authors to perform additional add-back experiments to formally demonstrate that the phenotype specifically relates to their KO. Moreover, while the single labeling strategy and multiple sampling time may be sufficient to obtain data on replication speed, I advise authors to also perform experiments using a standard dual-labeling approach, which might help them in identifying asymmetric forks that are a readout of fork stalling. The use of a dual labeling strategy might also be of help in the understanding of why loss of MCM8 or 9 does not affect replication progression at short labeling time while reducing greatly tract length with time. This is an interesting point that the authors should clarify.

Please see response to Reviewer 1, calculation of fork speed. Briefly, we have changed our wording to quantify 'replication rate' instead of 'fork speed'. We have also repeated this experiment using two other knockout clones (8B4 and 9G10) (Supplemental Figure 2) and can show a similar result. We agree that a dual labelling approach would be essential in calculating fork speed directly as well as identifying origin firing, however, we would like to reserve this experiment for a future manuscript examining redundant origin firing in the absence of MCM8 and 9.

Figure 3. In Fig2b authors do not show any difference in the level of DSBs between WT and KO cells in untreated conditions while in Fig3a-c they observe more H2AX+ cells but they state that the H2AX result is consistent with reduced fork progression. I suggest to specify the differences in the conclusion at the end of the paragraph.

In the results paragraph, we stated that this H2AX results is consistent with "defective replication that induces genomic stress", not just 'reduced fork progression'. We also discuss these differences for H2AX in the second paragraph of the Discussion (original lines 308-311), where H2AX also marks persistently stalled forks and single strand breaks which is consistent with the preventative role we are describing for MCM8/9. However, as the neutral comet assay (used here) detects DSBs but not other types of damage including single-strand breaks, it is likely the H2AX staining is more representative of the overall damage that includes rampantly reversed forks that 'represent' a DSB end, while Comet tails are DSB specific. We have added clarifying statements surrounding this discussion.

Figure 4. Authors state that, in untreated conditions and on HU, the IdU/CldU ratio in KO cells is reduced with respect to the WT and that may be related to fork instability, however, the relevant controls (i.e. nuclease KD or inhibition) are shown in Fig.8. They should be shown in Fig4 instead. Moreover, it would be useful to have an "internal" control, that is a condition known to induce fork degradation, to determine how strong is the phenotype conferred by MCM8/9 loss and if it is epistatic with known conditions leading to fork degradation (beside the experiments correlating the MCM8/9's role with that of BRCA1 reported in Fig6).

Thanks for this suggestion. We were concerned that having too many fiber results in one Figure would be overwhelming, however based on your suggestion we have rearranged the Figures accordingly (see the beginning of our response to you regarding rearrangement of Figures).

We are using WT (parental) cells as our internal negative control (new Fig 4b, plots 1&2) as well as siBRCA1 as our internal positive control (new Fig 6a, plots 1&2). Loss or knockdown of BRCA1, BRCA2, or Rad51 has been shown to correlate with a similar fork instability magnitude (see Ref 34 Taglialatela 2017) to what we show here. Moreover, in that same study, knockdown of SMARCAL1 restores fork integrity with HU in BRCA1, BRCA2, or Rad51 deficient cells also similar to what we show here with 8KO or 9KO cell lines.

Figure 5. Authors show that depletion of SMARCAL1 but not HTLF prevents signs of fork degradation in KO cells. First of all, I would recommend authors to also show data from mock-depleted points alongside data from depleted ones because this makes understanding of the data easier to readers. That said, it is needed to use an additional way to downregulate fork reversal to substantiate their claim.

We are using the WT parental cells as the internal control here for siRNA knockdown and fiber analysis, and as such, all cells as treated with the same siRNA. This is arguably a better control than mock depleting KO cell lines. Because we can show restoration of DNA fiber lengths upon siSMARCAL1 in all cell lines (as has been shown previously, again Ref 34) and can show a western blot of the protein depletion, we are confident in the controls and interpretations. Moreover, because the same restoration cannot be shown in siHTLF cell line, we note the importance in SMARCAL1 in reversing stalled forks in this MCM8/9 pathway.

Figure 6. Rationale of the experiments shown in Fig6 is completely unclear to me. Why do you expect BRCA1 loss mitigates the MCM8 or 8 KO? At least to me, It sounds like the hypothesis is that BRCA1 and MCM8/9 are epistatic or not. Alternatively, as data suggest, if loss of BRCA1 restores the WT phenotype, authors should provide experiments supporting a mechanism. For instance, does BRCA1 depletion prevent loading of exonucleases at deprotected forks in the absence of MCM8/9?

This is also related to Reviewer 1, comment 3.

This is complex and somewhat enigmatic for sure. siBRCA1 makes parental cells more susceptible to degradation in the presence of HU because the forks are unprotected (as also seen in Ref 35). However, initiation of this reversal/protection pathway must also be dependent on MCM8/9. We hypothesize that MCM8/9 are early effectors in the fork reversal/protection pathway, and so, when MCM8 or 9 are absent and BRCA1 is knocked down, there appears to be limited fork reversal susceptible to nucleases. In fact, MCM8/9 may be a gatekeeper for the fork progression/reversal-protection pathways as supported by our BRCv and WA results in Fig 7. As much as we would like to show a mechanism, it is complicated by the somewhat poorly understood role of BRCA1 in comparison to BRCA2. For now, we can show this switch (and dependence) from MCM8/9 to Rad51 and BRCA1 to stabilize stalled forks, but further mechanistic studies are still ongoing.

How do the authors reconcile the role of BRCA1 in protecting forks with the apparent reduction in BRCA1 foci in response to HU when MCM8/9 are KO?

Related to the response above, but without a better understanding of the exact role of BRCA1, this is hard to reconcile. However, it appears that MCM8/9 act as a switch to activate the controlled handoff to Rad51 and BRCA1 for stalled fork reversal and protection. Normally,

MCM8/9 will promote fork progression, but once stalled, MCM8/9 is needed to initiate this pathway switch towards reversed fork protection. When MCM8/9 is absent and BRCA1 is present (Fig 4b, plots 3&5 and 7&9, and also represented by the panels in Fig 6b without GFP) there is rampant fork reversal and limited protection. When MCM8/9 is transfected (likely at high concentrations - Fig 6b, with GFP) then it is stimulating a fork progression conformation, where BRCA1 is not present. It is likely that MCM8/9 is removed once BRCA1 acts to stabilize stalled forks, thus the ying and yang of these two proteins in the nucleus. The increase in BRCA1 foci when MCM8 or MCM9 is knocked out represents stalled forks that are uncontrolled and aberrantly reversed and susceptible to nucleases.

This is a crucial point and the presented data are not fully supportive of the proposed difference between unstressed vs. stressed.

Again, it is our view that during unstressed replication, MCM8/9 act to facilitate forward fork progression, but during times of prolonged replication stress, MCM8/9 facilitate a controlled hand off to Rad51 and BRCA1 to initiate reversed fork protection.

We understand that we did not make these points entirely clear, and so we have updated the text in the results around Fig 6a with this sentence. “....., as it is out hypothesis the MCM8/9 is required to facilitate this pathway switch from fork progression to reversal/protection.”

And here in the results for Fig 6b. “Therefore, MCM8/9 likely acts to antagonize BRCA1-mediated fork processing/stabilization during fork reversal to maintain replication fork stability during normal replisome progression. However, when severe replisome stalls are prevalent, MCM8/9 hands off the fork template for controlled reversal/protection through the BRCA1/2, Rad51, SMARCAL1 nexus, essentially swapping control of the template.”

We have also updated the text in the discussion and included consistent language (i.e. Fork Progression, Protection, Degradation, and Cleavage in an updated Fig 8a, b, and c) to direct the reader appropriately. We hope this now more clearly described.

As suggested above, it absolutely needed to show mock-depleted data in the graphs to avoid continuous back and forth through figures.

As discussed above, we have utilized the parental strain throughout as an appropriate control instead of mock-depletion in KO cell lines. We argue that this arrangement provides sufficient matched controls to examine and relate the various DNA fiber experiments.

However, we have also updated the Figures as suggested to keep the relevant data together, and when the data is in separate figures, such as for Fig 6a, we have now added text to direct the reader to the specific appropriate plots for comparison.

Figure 7. Authors performed experiments to determine if the BRCv motif or the helicase activity of MCM9 was involved in fork stabilization. This is an important point that would require previous analysis of whether MCM9 (or MCM8) destabilizes RAD51 filaments at the reversed fork.

The dependence of Rad51 on MCM8/9 was shown in our previous paper (Ref 30), where we showed a significant decrease (almost an absence) of Rad51 foci when MCM8 or MCM9 was

knockout out or in mutated in patient cell lines. We do not know yet whether MCM8/9 actively destabilizes Rad51 filaments (which would require in vitro biochemical studies we are working on) or just prevents Rad51 from having a controlled deposition onto a stalled fork.

Figure 8. This figure should be merged with Fig4. Authors should include mock-depleted data in the graphs. Authors show that basically each exonuclease is apparently involved in degrading reversed forks in KO cells. Usually, destabilized forks get degraded by MRE11-EXO1 while DNA2 is mostly involved in the limited resection thought to take place in WT setting. How do authors reconcile with this? Little explanation is provided in discussion.

We chose to merge this old Fig. 8 with Fig. 5 to show fork reversal, protection, and cleavage in one combined figure (new Fig 5). We felt it would be too much to also include this in Fig 4 as we updated that one to show nascent strand degradation is primarily from MRE11.

As far as separation of function for these nucleases, this is an ongoing complex issue and was also mentioned by Reviewer 1. Please see our response there. However, we recognize that we were not specific enough in our original submission regarding the results from the various nucleases. We have added more clarifying statements in the Results section surrounding Fig 4cf that are supported by newly added references. Briefly, we agree with this reviewer, that MRE11-EXO1 are degrading the majority of destabilized forks in the absence of MCM8/9, while DNA2 has a minor fork processing role, likely with substrates that are discreetly different but still yet uncharacterized.

Minor comments:

A general suggestion is to revise wording and structure of some sentences in the text and in the legends. I found some redundancy here and there.

- I think the introduction might be improved by avoiding some redundancies existing when talking about previous evidence of an MCM8/9 role during replication

We have removed some of the redundancies and streamlined the intro, however, these previous results regarding a role for MCM8/9 in replication are also important to include in light of our results.

- I found few typos throughout the manuscript (e.g. line 123 “genomic instability” where it should read “genomic stability”

Fixed, thank you.

- Some figures can be merged also to make catching the result easier (e.g. Fig2 + Fig3; Fig4+Fig5) or moved as supplemental information (e.g. Fig.2a can be put in the supplemental material)

We have merged several of the figures and added several other supplemental experiments and figures to strengthen and streamline the manuscript as suggested. This are listed above.

- Line 149. The first sentence of the paragraph is an overstatement. The previous paragraph

just shows that KO cells replicate slower when unchallenged and have or not have depending on the assay more DNA damage.

Yes ok, maybe too soon. We have reworded this sentence to be a better introduction into the DNA fiber experiments to examine fork instability without overstating the current findings.

- Titles of some figures are not consistent with the key finding.

We have double checked and updated the Figure Legend Titles to better represent the key findings.

REVIEWER COMMENTS

Reviewer #1 (Remarks to the Author):

In the revised manuscript, Griffin et al have clarified the role of MCM8/9 at the replication fork in regards to regulation of fork progression and fork reversal by MCM8/9. The new additions to the results and discussion and modification of the model improve the manuscript and help to clarify the relationship between MCM8/9 and BRCA1/BRCA2/RAD51. This, combined with the inclusion of additional clones and information about number of replicates make this a strong manuscript that describes a new role for MCM8/9 at the replication fork in protecting nascent strands from degradation by regulating fork progression and reversal. This manuscript will likely be of broad interest. The authors have addressed my concerns, and I believe the manuscript is sufficient for publication.

Reviewer #2 (Remarks to the Author):

Previously, I raised multiple experimental issues. In the revised manuscript, the authors added new data. To exclude the possibility of clonal variation, they checked other KO clones (Figures S1, 7) and carried out rescue experiments (Figure S2). They also looked at transfection efficiency used for the rescue assays showing it was about 80% (Figure S3). Even though the authors added these data, I still have some concerns about the same issues. I wish the authors would address the following points.

The MCM8KO and 9KO lines

I understand that these cell lines were initially described in their previous publication (Figure S4 in McKinze et al. JCB, 2021). However, the authors have not shown the immunoblot data, even though I suggested confirming the knockouts by immunoblotting (this is relevant to the issue raised by Reviewer 1). Although the authors said that it was difficult to detect the MCM8 and 9 proteins by immunoblot because of the low expression levels, these proteins were clearly detected in other publications (Lutzmann et al. Mol Cell, 2008, 2012; Nishimura et al. Mol Cell, 2012; Park et al. MCB, 2013; Lee et al. Nat Commun., 2015). It is essential to confirm that an aberrant form of MCM8 or 9 is not expressed in the knockouts.

Rescue experiments

I noted that the pEGFP plasmid used for the rescue assays contains the SV40 origin. Therefore, it is replicated in 293T cells causing massive overexpression. Even though it is difficult, it is important to evaluate the overexpression level compared to the endogenous MCM8 and 9 levels.

In some rescue experiments, such as Figure S2 and Figure 4b (plot 6), the phenotype was completely rescued even though the transfection efficiency was 80%. Can the authors explain why?

Calculation of fork speed

As the authors admitted in the rebuttal, it is impossible to evaluate the fork speed using single labelling (Figure 1a). The revised version said 'replication rate' instead of fork speed in the text. However, replication rate is the sum of origin firing and fork progression, and the authors focus on 'replication fork progression' (line 118) after Figure 1. To make a logical connection in the story, it is crucial to evaluate fork speed using double labelling.

Providing appropriate control

The authors rearranged figures showing nascent-strand degradation (NSD) in Figures 4, 5, and 6. However, enough control data were not presented yet. All siRNA knockdown data do not show control siRNA treatment (Figures 4c, e, f, 5a, b, d and 6a). The transfection procedure in these experiments was slightly different from the one used in Figure 4b. Thus, Figure 4b cannot be a control for the siRNA experiments. In Figures 4d and 5c, there is no Mirin treatment control. In Figure 7, there are no

WT control cells.

Mechanistic insights

One of the important mechanistic insights into how MCM9 works at the stalled replication fork is RAD51 loading by MCM9, based on the data shown in Figure 7. The authors used the MCM9 BRCv mutant, which lacked the RAD51 interaction domain. This interaction was previously demonstrated using Y2H and pull-down assay with purified proteins by the same group (Figure 8 in McKinze et al. JBC 2021). However, it is still unclear whether this interaction occurs in the living cells. Therefore, it is crucial to show the interaction between MCM9 and RAD51 by IP or PLA in cells stressed by HU and confirm it is BRCv dependent.

Other issues

"To examine this possibility, we measured replication fork stability in WT, 8KO, and 9KO cells by DNA fiber analysis (Fig. 4a)." (line 154)

The authors said 'replication fork stability' without explaining what they were looking at. Replication fork stability is vague because this can be the protection from nascent-strand degradation or double-strand break at the stalled fork. The authors should explain what they looked at and cite appropriate references about the assay (such as Schlacher et al. Cell, 2011).

Reviewer #3 (Remarks to the Author):

I carefully read the revised version of the manuscript and the point-by-point replies provided to the reviewers.

Collectively, I found the manuscript improved and I appreciated the effort of the authors to deal with the comments made by reviewers.

In my opinion, the revisions addressed in a fairly satisfactory way my comments and the previously-identified weaknesses.

I am not fully convinced by the proposed models, however, but I agree that it is difficult to depict exactly the relationship between MCM8/9, B1 and FR enzymes. Thus, I agree that the explanations offered by the authors are sufficient to support the identification of a novel role for MCM8/9 at the RFs.

I have two minor suggestions that authors may or may not address.

1. Line 243, in the results, indicates the role of MUS81 in recovering forks under pathological conditions but does not contain any citation to the pathological condition (e.g. loss of WRN, oncogene-induced replication stress, B2 deficiency and others). I think that authors should consider including citations to the most important seminal works here.
2. Line 397, in the discussion, the authors offer an explanation of the role of MCM8/9 that echoes the one offered to reviewers in their rebuttal letter. In the manuscript, the statement on a hypothetical role of MCM8/9 as regulator of fork reversal is toned down correctly. I appreciated this. However, I was arguing whether the authors might refer in their discussion to other conditions resembling how MCM8/9 might act. For instance, the dual role of MCM8/9 during unperturbed vs. perturbed replication is somehow reminiscent of what has been proposed for RADX and a role as "emergency break" for fork reversal is somehow reminiscent of the "gatekeeper" role of RAD52. Discussion of common or MCM8/9 KO-specific phenotypes might contribute to put their discoveries in the general context of transactions at forks and highlight the peculiar role of MCM8/9.

Response to Reviewers

We so glad we have addressed all of Reviewer #1 and #3 concerns and they believe it is suitable for publication, and we thank all the Reviewers for their comments to help make this a much stronger manuscript.

Any further changes in the manuscript and supplementary information are indicated as red text as related manuscript files.

Reviewer #1 (Remarks to the Author):

In the revised manuscript, Griffin et al have clarified the role of MCM8/9 at the replication fork in regards to regulation of fork progression and fork reversal by MCM8/9. The new additions to the results and discussion and modification of the model improve the manuscript and help to clarify the relationship between MCM8/9 and BRCA1/BRCA2/RAD51. This, combined with the inclusion of additional clones and information about number of replicates make this a strong manuscript that describes a new role for MCM8/9 at the replication fork in protecting nascent strands from degradation by regulating fork progression and reversal. This manuscript will likely be of broad interest. The authors have addressed my concerns, and I believe the manuscript is sufficient for publication.

Thank you for your comments!

Reviewer #2 (Remarks to the Author):

Previously, I raised multiple experimental issues. In the revised manuscript, the authors added new data. To exclude the possibility of clonal variation, they checked other KO clones (Figures S1, 7) and carried out rescue experiments (Figure S2). They also looked at transfection efficiency used for the rescue assays showing it was about 80% (Figure S3). Even though the authors added these data, I still have some concerns about the same issues. I wish the authors would address the following points.

The MCM8KO and 9KO lines

I understand that these cell lines were initially described in their previous publication (Figure S4 in McKinzey et al. JCB, 2021). However, the authors have not shown the immunoblot data, even though I suggested confirming the knockouts by immunoblotting (this is relevant to the issue raised by Reviewer 1). Although the authors said that it was difficult to detect the MCM8 and 9 proteins by immunoblot because of the low expression levels, these proteins were clearly detected in other publications (Lutzmann et al. Mol Cell, 2008, 2012; Nishimura et al. Mol Cell, 2012; Park et al. MCB, 2013; Lee et al. Nat Commun., 2015). It is essential to confirm that an aberrant form of MCM8 or 9 is not expressed in the knockouts.

We were able to recently obtain newer commercial antibodies that allowed us to better visualize endogenous levels of MCM8 and MCM9 by Western blot and confirm the knockouts (or commensurate knockdowns) for MCM8 or MCM9 in all three cell lines (now shown in Fig 1a-b). MCM8 and MCM9 are effectively absent from the respective knockout cell lines and removal of one protein has effects on the expression levels of the other. We have updated the text describing these results in comparison to previous reports.

Rescue experiments

I noted that the pEGFP plasmid used for the rescue assays contains the SV40 origin. Therefore, it is replicated in 293T cells causing massive overexpression. Even though it is difficult, it is important to evaluate the overexpression level compared to the endogenous MCM8 and 9 levels.

We tried to explain this in our previous rebuttal, where we provided a western blot of this overexpression, and we do not disagree that there will be massive overexpression of MCM8 or MCM9 from a SV40 origin and CMV promoter. However, this is not dissimilar to what most all other researchers do when adding back function from an exogenous plasmid, and without an exact method to control the overexpression levels and another method to validate those levels to compare with the current results, this does not seem relevant or feasible. Except to say that any amount of expression of MCM8 or 9 will be well above any endogenous levels. Moreover, we are only overexpressing (i.e. rescuing) either MCM8 or MCM9 and not both at the same time, and so, any significant excess of one protein will likely not be stable without a commensurate level of the other.

We have changed the text to read "...are rescued by the overexpression of untagged MCM8 or MCM9..."

In some rescue experiments, such as Figure S2 and Figure 4b (plot 6), the phenotype was completely rescued even though the transfection efficiency was 80%. Can the authors explain why?

In both these cases, the median ratios appear to be fully rescued, however, the distributions are still a bit different. In Fig 4b (plot 6 and also plot 10 with +MCM9), there are several longer ratios but also broader distributions within both the upper and lower quartile ranges compared to plot 2 in that same panel 4b. Therefore, while rescue is happening, there are also fibers from cells in the lower quartile that are not effectively rescued, likely from the <100% transfection efficiency as you mention. There are also cells in the upper quartile that have longer IdU stretches for which we do not yet have suitable explanation. We have added a new Supplemental Figure 8 that examines the statistical distributions for all the fiber ratios in Figure 4-6, noting the quartiles and median for the control condition compared to the test condition as discussed in the text. We hope this better illustrates the complexity of the distributions, and that although we are effectively rescuing MCM8 or 9 knockout lines, the distributions are not perfect based on the experimental caveats.

We have also reanalyzed that data from Fig S2 (now Fig S3) and see a very similar result as previous, that rescue was complete. We have included violin plots in this figure to include more statistics. It appears that at the longest time point, WT cells have longer lengths than either 8KO +MCM8 or 9KO +MCM9 suggesting less than 100% restoration. Of course, we are cognizant of the error associated with single pulse experiments, especially at longer time points, as you pointed out, and so, we have now included dual label experiments to calculate fork speed as you have suggested in Fig 1e-g (see next comment).

Calculation of fork speed

As the authors admitted in the rebuttal, it is impossible to evaluate the fork speed using single labelling (Figure 1a). The revised version said 'replication rate' instead of fork speed in the text. However, replication rate is the sum of origin firing and fork progression, and the authors focus on 'replication fork progression' (line 118) after Figure 1. To make a logical connection in the story, it is crucial to evaluate fork speed using double labelling.

We have now performed dual labelled fork progression experiments with a single 30-minute CldU pulse followed by 30, 45, and 60 min IdU pulses to specifically examine only fork progression (Fig. 1e-g) and eliminate any conflicting analysis from additional origin firing or other repair associated issues. The results are interesting in that there is not a significant increase in IdU lengths after 30 minutes in the knockout cell lines, suggesting that a combination severe fork stalling, nascent strand degradation, and breaks are occurring during the second pulse as we detail throughout the rest of the manuscript. So, it is likely that this reviewer is correct that a combination of new origin activation was compromising our single pulse data (now moved to Supplemental Figs 1-3), where we saw a slight increase (but significantly reduced from WT) in lengths over that time. We think the combination of both the single and dual pulses provides a more complete story of the role of MCM8/9 in promoting fork progression. The investigation of new origin firing when MCM8/9 is absent is a strong possibility and will be a focus for the future but will require significant further experimental development to examine the spectrum of genomic survival outcomes when MCM8 or 9 are absent.

Providing appropriate control

The authors rearranged figures showing nascent-strand degradation (NSD) in Figures 4, 5, and 6. However, enough control data were not presented yet. All siRNA knockdown data do not show control siRNA treatment (Figures 4c, e, f, 5a, b, d and 6a). The transfection procedure in these experiments was slightly different from the one used in Figure 4b. Thus, Figure 4b cannot be a control for the siRNA experiments. In Figures 4d and 5c, there is no Mirin treatment control. In Figure 7, there are no WT control cells.

Control siRNA treatment is a great control if you are examining a single cell line and comparing siRNA knockdown to that of a transfection control alone. However, in all cases throughout our manuscript, we are using the parental cell lines as the control, where they are treated and/or knocked down identical to that of the test MCM8 or MCM9 knockout cell lines. The comparison of the test case is always back to the parental cell line or +/- various identical treatments. So, we actually have a more thorough cellular and transfection control compared with just a control siRNA for a single cell line. We suggest that adding this control siRNA experiment is not only redundant but is actually an inferior control to what we have performed. Moreover, we feel that adding additional siRNA control knockdowns for each situation would unnecessarily complicate our current plots, where there are already many different conditions to examine.

Instead, to better compare the results described in the text, we have now included violin plots (Fig 1f and Supp Fig S8) for the various specific comparisons that we discuss in the results. We hope this provides the reader an alternative way to visualize and interpret a subset of the data.

Mechanistic insights

One of the important mechanistic insights into how MCM9 works at the stalled replication fork is RAD51 loading by MCM9, based on the data shown in Figure 7. The authors used the MCM9 BRCv mutant, which lacked the RAD51 interaction domain. This interaction was previously demonstrated using Y2H and pull-down assay with purified proteins by the same group (Figure 8 in McKinze et al. JBC 2021). However, it is still unclear whether this interaction occurs in the living cells. Therefore, it is crucial to show the interaction between MCM9 and RAD51 by IP or PLA in cells stressed by HU and confirm it is BRCv dependent.

In our previous publication (McKinze et al. JBC 2021), we were able to show an interaction between MCM9 and RAD51 through several complementary assays. Moreover, mutation of this BRCv site in MCM9 disrupted this physical interaction (Fig 8 in that paper), and complementary experiments showed knockout of MCM9 reduced (almost eliminated) RAD51 foci in cells (Fig 7D in that paper), mutation of the MCM9 BRCv site eliminate damage induced MCM9 foci (Fig 5D in that paper), and there is partial colocalization of RAD51 with MCM9 (Fig 6B in that paper) were ALL MCM9 foci were colocalized with RAD51 (w/ many more individual Rad51 foci).

Park et al 2013 already showed that an IP for MCM9 (+/- cisPt) also pulled down RAD51. However, we are also suggesting that while this MCM9(BRCv) motif is important for RAD51 recruitment, it is likely that this interaction is not stable. This is based on the weaker IP seen in the Park et al 2013 paper and incomplete localization in our JBC 2021 paper. Therefore, we have changed any wording in this manuscript to be more careful in describing a dependence of the MCM9-BRCv motif for RAD51 recruitment but have eliminated any mention of a stable interaction.

Other issues

"To examine this possibility, we measured replication fork stability in WT, 8KO, and 9KO cells by DNA fiber analysis (Fig. 4a)." (line 154)

The authors said 'replication fork stability' without explaining what they were looking at. Replication fork stability is vague because this can be the protection from nascent-strand degradation or double-strand break at the stalled fork. The authors should explain what they looked at and cite appropriate references about the assay (such as Schlacher et al. Cell, 2011).

Agreed, we could have been more specific in our resubmission. We have added the suggested reference and have been more careful in utilizing nascent strand degradation (NSD) and fork protection throughout where appropriate, instead of just replication fork stability, which is more vague.

Reviewer #3 (Remarks to the Author):

I carefully read the revised version of the manuscript and the point-by-point replies provided to the reviewers.

Collectively, I found the manuscript improved and I appreciated the effort of the authors to deal with the comments made by reviewers.

In my opinion, the revisions addressed in a fairly satisfactory way my comments and the

previously-identified weaknesses.

I am not fully convinced by the proposed models, however, but I agree that it is difficult to depict exactly the relationship between MCM8/9, B1 and FR enzymes. Thus, I agree that the explanations offered by the authors are sufficient to support the identification of a novel role for MCM8/9 at the RFs.

We are glad that we have addressed most of your concerns in this past revision.

I have two minor suggestions that authors may or may not address.

1. Line 243, in the results, indicates the role of MUS81 in recovering forks under pathological conditions but does not contain any citation to the pathological condition (e.g. loss of WRN, oncogene-induced replication stress, B2 deficiency and others). I think that authors should consider including citations to the most important seminal works here.

Our omission and references should have been included. We have now included several references (44-46) describing this effect.

2. Line 397, in the discussion, the authors offer an explanation of the role of MCM8/9 that echoes the one offered to reviewers in their rebuttal letter. In the manuscript, the statement on a hypothetical role of MCM8/9 as regulator of fork reversal is toned down correctly. I appreciated this.

However, I was arguing whether the authors might refer in their discussion to other conditions resembling how MCM8/9 might act. For instance, the dual role of MCM8/9 during unperturbed vs. perturbed replication is somehow reminiscent of what has been proposed for RADX and a role as "emergency break" for fork reversal is somehow reminiscent of the "gatekeeper" role of RAD52. Discussion of common or MCM8/9 KO-specific phenotypes might contribute to put their discoveries in the general context of transactions at forks and highlight the peculiar role of MCM8/9.

We would love to be able to put MCM8/9 in context with RADX and/or RAD52 among others including HROB, and of course I could speculate, but I feel this would just be too preliminary for this manuscript. Of course, this work is ongoing, and we hope to have a more suitable explanation or differentiation in a forthcoming manuscript.

REVIEWERS' COMMENTS

Reviewer #2 (Remarks to the Author):

I found the revised manuscript was significantly improved. I appreciate the effort in carrying out all experiments and revising the manuscript. Even though I am not fully satisfied with the authors' explanation of the siRNA control issue, I understand that it would be challenging to repeat all NSD assays including the control. Now I feel the described roles of MCM8/9 in fork protection are ready to be shared and then confirmed by the research community.

Before moving forward, I have a minor issue.

"It was previously shown in mice that the stability of MCM8 and MCM9 were dependent on each other, as knockout or knockdown of one also reduced or eliminated levels of the other (19)" (lines 6 to 8 from the bottom, page 6).

The interdependency of MCM8 and MCM9 in chicken and human cells was also reported (Nishimura et al. Mol Cell, 2012; Park et al. MCB, 2013).

Response to Reviewers and Editorial Queries.

Thanks for accepting this manuscript (in principle). We have made some subtle changes (noted in red) that were in response to Rev 2 and your Editorial Requests.

Added new Supplementary Figure 1 to show the gating for the FACS experiments in Fig 1b-c and then had to adjust all the other figures numbering down by 1.

Reviewer #2 (Remarks to the Author):

I found the revised manuscript was significantly improved. I appreciate the effort in carrying out all experiments and revising the manuscript. Even though I am not fully satisfied with the authors' explanation of the siRNA control issue, I understand that it would be challenging to repeat all NSD assays including the control. Now I feel the described roles of MCM8/9 in fork protection are ready to be shared and then confirmed by the research community.

Before moving forward, I have a minor issue.

"It was previously shown in mice that the stability of MCM8 and MCM9 were dependent on each other, as knockout or knockdown of one also reduced or eliminated levels of the other (19)" (lines 6 to 8 from the bottom, page 6).

The interdependency of MCM8 and MCM9 in chicken and human cells was also reported (Nishimura et al. Mol Cell, 2012; Park et al. MCB, 2013).

We have removed the words 'in mice' and added the suggested references to indicate interdependency across species.